# Universal growth of perovskite thin monocrystals from high solute flux for sensitive self-driven X-ray detection

Da Liu[1], Yichu Zheng[2], Xin Yuan Sui[1], Xue Feng Wu[1], Can Zou[1], Yu Peng [1], Xinyi Liu[1], Miaoyu Lin[1], Zhanpeng Wei[1], Hang Zhou[3], Ye-Feng Yao [3], Sheng Dai [4], Haiyang Yuan[1], Hua Gui Yang [1], Shuang Yang [1]✉ & Yu Hou [1]✉

Metal-halide perovskite thin monocrystals featuring efficient carrier collection and transport capabilities are well suited for radiation detectors, yet their growth in a generic, well-controlled manner remains challenging. Here, we reveal that mass transfer is one major limiting factor during solution growth of perovskite thin monocrystals. A general approach is developed to overcome synthetic limitation by using a high solute flux system, in which mass diffusion coefficient is improved from $1.7 \times 10^{-10}$ to $5.4 \times 10^{-10}$ $m^2$ $s^{-1}$ by suppressing monomer aggregation. The generality of this approach is validated by the synthesis of 29 types of perovskite thin monocrystals at 40–90 °C with the growth velocity up to 27.2 μm $min^{-1}$. The as-grown perovskite monocrystals deliver a high X-ray sensitivity of $1.74 \times 10^5$ μC $Gy^{-1}$ $cm^{-2}$ without applied bias. The findings regarding limited mass transfer and high-flux crystallization are crucial towards advancing the preparation and application of perovskite thin monocrystals.

Radiation detectors that directly convert X-ray photons into an electrical signal are essential in various fields such as medical imaging, homeland security, astrophysics, and scientific research applications[1-3]. In recent years, self-driven X-ray detectors have emerged without the requirement of an outside power source, providing reliable technological solutions for portable, remote areas and harsh space applications[4]. However, progress in this field has been limited primarily by the trade-off between photon attenuation and charge collection efficiency[5]. For instance, millimeter-thick crystals of widely applied Si are needed to attenuate 50 keV X-ray photons[6], which obviously exceeds their carrier diffusion length and cannot offer detectable signals under zero bias[7].

Metal-halide perovskite monocrystals featuring high mobility-lifetime products (up to $10^{-3}$ $cm^2$ $V^{-1}$) and large X-ray attenuation coefficients (up to 40 $cm^{-1}$ for 50 keV X-ray photons) are promising candidates for the development of radiation detectors[8,9]. Further thinning the monocrystal to a thickness of less than 100 μm enables both efficient electronic charge collection and optical attenuation[10], which offers an opportunity for the high-performance self-driven X-ray detector. The solution growth of perovskite thin monocrystals is generally undergone at confined space[11], liquid-air interface[12], or epitaxial surface[13], driven by the solute supersaturation upon inverse temperature effect[14] or solvent evaporation[15]. Up to now, many non-lead and multi-cation/anion perovskite thin monocrystals are still difficult to synthesize via a generic, well-controlled manner. In addition, the development of perovskite thin monocrystals has been limited primarily by their time-consuming preparation and inferior crystalline quality. For example, the growth of $MAPbI_3$ thin monocrystal takes approximately about 4 days and results in a relatively high trap density,

[1]Key Laboratory for Ultrafine Materials of Ministry of Education, Shanghai Engineering Research Center of Hierarchical Nanomaterials, School of Materials Science and Engineering, East China University of Science and Technology, 130 Meilong Road, 200237 Shanghai, China. [2]School of Mechatronic Engineering and Automation, Shanghai University, 99 Shangda Road, 200444 Shanghai, China. [3]Physics Department & Shanghai Key Laboratory of Magnetic Resonance, School of Physics and Electronic Science, East China Normal University, 3663 North Zhongshan Road, 200062 Shanghai, China. [4]Key Laboratory for Advanced Materials and Feringa Nobel Prize Scientist Joint Research Center, Institute of Fine Chemicals, School of Chemistry & Molecular Engineering, East China University of Science and Technology, 130 Meilong Road, 200237 Shanghai, China. ✉e-mail: syang@ecust.edu.cn; yhou@ecust.edu.cn

exceeding $10^{13}$ cm$^{-3}$ ($-10^{10}$ cm$^{-3}$ for bulk monocrystal)[16,17]. Previous research has mainly focused on controlling the nucleation kinetics by ligand molecules[18,19], low-temperature gradient[20], or liquid diffusion[21], which do not remarkably contribute to the following growth step. In essence, crystal growth is undoubtedly related to the mass transfer and surface reaction velocities, in which the slower one dominates the growth kinetics[22]. Concerning the fast surface condensation of monomers during crystal growth, mass transfer, as a ubiquitous process, has not yet been well understood, which is closely linked with the growth rate and defect formation of perovskite thin monocrystals[23].

In this work, we show that mass transfer, which has been rarely discussed in perovskite growth, limits the overall formation of perovskite thin monocrystals. We design a growth strategy based on high solute flux to overcome this limitation, enabling the universal growth of a library of perovskite thin monocrystals via a low-temperature (<90 °C), rapid (up to 27.2 μm min$^{-1}$) manner. The fast mass transfer guarantees the uniform supply of precursors and suppresses the bulk defect formation of perovskite monocrystals. As a consequence, the obtained Cs$_{0.02}$FA$_{0.2}$MA$_{0.78}$PbI$_3$ thin monocrystals (where MA is methylamine, FA is formamidine) exhibit a high mobility-lifetime product of $2.53 \times 10^{-3}$ cm$^2$ V$^{-1}$ and a long carrier diffusion length of 82.7 μm. The self-driven detector based on the as-grown thin monocrystal enables X-ray imaging and attains an impressive X-ray sensitivity of $1.74 \times 10^5$ μC Gy$^{-1}$ cm$^{-2}$ and the lowest detection limit of 11.8 nGy s$^{-1}$.

## Results

### Growth and characterization of perovskite thin monocrystals

The MAPbI$_3$ thin monocrystals are synthesized via the space-confined growth method, which typically incorporates nucleation and growth processes. For hybrid perovskites, the growth process usually consumes solutes rapidly and requires a high solute flux for the surface reaction. The insufficient solute supply may permit the adsorption of adatom on the crystal surface, other than along the stable terrace or kink, resulting in undesired nucleation or imbalanced growth[24]. The flat, hydrophobic substrates are adopted to minimize the nucleation density (Supplementary Figs. 1 and 2). To facilitate the mass transfer, we selected glycol ethers, e.g., 2-methoxyethanol (2-ME), as solvents with several potential advantages, including fast solute diffusion, high solubility (2.5 mol L$^{-1}$ for MAPbI$_3$), and large inverse temperature solubility between 25–70 °C (Supplementary Fig. 3), which allows a high-flux growth (abbreviated as HFG) for MAPbI$_3$ thin monocrystal growth. The solution was heated to 70 °C with a heating rate of 4 °C h$^{-1}$ (Supplementary Fig. 4). In case of the formation of excess nucleus, we employed a solution with slightly lower than the up limit (2 mol L$^{-1}$) to growth MAPbI$_3$ monocrystals (Supplementary Fig. 5 and Note 1). Upon one growth cycle of 48 h, the length of MAPbI$_3$ thin monocrystal approaches 2.0 cm (Fig. 1a). In comparison, thin monocrystals from commonly used γ-butyrolactone (GBL) solvent experience slow growth velocity at high temperature (abbreviated as control). The solution was heated from 60 to 130 °C with a ramp rate of 2 °C h$^{-1}$ and the crystal size reached ~3 mm after 96 h (Supplementary Fig. 6)[20].

X-ray diffraction (XRD) spectrum of the HFG thin crystal shows strong diffraction peaks of (200) and (400) planes at 20.03° and 40.70° (Fig. 1a). The diffraction intensity of HFG crystal is almost twice as high as that of the control crystal (Supplementary Fig. 7), indicating the high crystallinity. The HFG thin monocrystal exhibits the full width at half maximum (FWHM) of only 0.016° from a high-resolution X-ray rocking curve (Fig. 1b), which is comparable to that of the bulk MAPbI$_3$

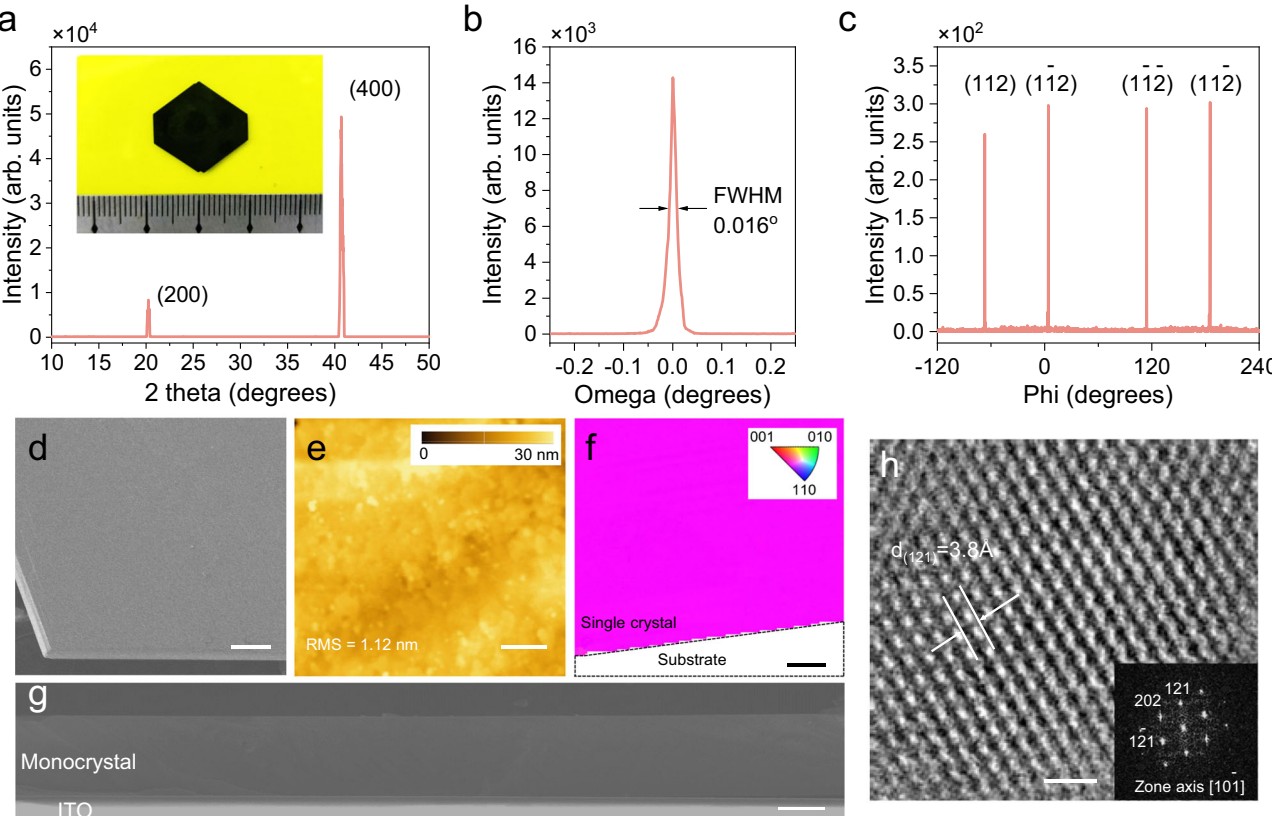

**Fig. 1 | Structural characterizations of MAPbI$_3$ thin monocrystals. a** XRD pattern of thin monocrystal from high-flux growth (HFG). The inset shows the photograph of the HFG thin monocrystal. **b** Rocking curve of the (400) diffraction peak of the HFG thin monocrystal. **c** Phi scan curve of the (112) diffraction peak of the HFG thin monocrystal. **d** Top-view scanning electron microscope (SEM) images of the HFG thin monocrystal. Scale bars: 100 μm. **e** AFM height image of the HFG thin monocrystal. Scale bars: 1 μm. **f** EBSD inverse pole figure mapping of the top surface of the HFG thin monocrystal. Scale bars: 100 μm. **g** Cross-sectional SEM image of the HFG thin monocrystals. Scale bars: 30 μm. **h** HRTEM image and the corresponding fast Fourier-transform (FFT) pattern of the HFG thin monocrystal. Scale bars: 1 nm.

monocrystal[19]. The presence of four peaks at an interval of 109° and 71° in the $\varphi$ scan fixed along MAPbI$_3$ <112> in Fig. 1c unambiguously confirms its monocrystalline eminence. The pole figure in Supplementary Fig. 8 sketches four discrete diffraction spots at $\varphi$ = 54.5°, 125.5°, 234.5°, and 305.5°, in good consistency with the $\varphi$ scan curves. Notably, the spots are not separated azimuthally by 90°, but have a certain diffraction orientation relationship, which corresponds to the simulated pole figure of MAPbI$_3$. These combined X-ray diffraction results confirm the high crystallinity and reduced misorientations of the thin monocrystal synthesized from high solute flux.

As shown in Fig. 1d, the HFG thin monocrystal shows sharp edges and a well-defined morphology. The surface quality of the thin monocrystal is evaluated by the atomic force microscope (AFM). The HFG crystal demonstrates an ultra-smooth surface with a root-mean-square roughness of 1.12 nm (Fig. 1e), which is much lower than that of the control crystal (4.42 nm, Supplementary Fig. 9). We employed electron backscatter diffraction (EBSD) technique to examine the phase purity of the HFG thin monocrystal (Fig. 1f). The Z axis inverse pole figure presents the uniform color, that is, individual crystallographic orientation[25]. The thickness of the thin monocrystal is about 50 μm (Fig. 1g), corresponding to a high aspect ratio of 379 for MAPbI$_3$ perovskite. The crystal thickness can also be controlled in a wide range between 1–60 μm by applying loads (Supplementary Fig. 10). High-resolution transmission electron microscopy (HRTEM) image in Fig. 1h presents the distinct lattice plane fringes with $d$-spacing of 0.38 nm, assigned to the (121) lattice plane of MAPbI$_3$. A fast Fourier-transformed (FFT) image exhibits distinct spot patterns consistent with the simulated ones of MAPbI$_3$ tetragonal phase ($I4\,cm$ space group, Supplementary Fig. 11)[26]. Energy dispersive X-ray spectroscopy (EDX) mappings of C, N, Pb, and I further confirm the elemental uniformity of MAPbI$_3$ thin monocrystals (Supplementary Fig. 12).

## Crystal growth mechanism

The space-confined growth of thin monocrystals can be described by a two-step growth model (Fig. 2a), which consists of the solute diffusion and the condensation of monomers on the crystal surface[27]. The accessibility of monomers at the crystal surface determines the crystal growth velocity[28]. We established a theoretical crystal growth model based on MAPbI$_3$ monocrystal to explore the solute distribution (Supplementary Note 2). The diffusion coefficient ($D$) is assumed to be $0.5 \times 10^{-10}$–$4 \times 10^{-10}$ m$^2$ s$^{-1}$, which is comparable to ionic diffusion in liquids, like CaCl$_2$ in water at 25 °C ($7.78 \times 10^{-10}$ m$^2$ s$^{-1}$)[29]. The variable $x$ is defined as the distance to the surface of the monocrystal. It is rather surprising that the solute concentration is decreased from 1.5 mol L$^{-1}$ of $C_b$ ($x = \infty$) to 0.8 mol L$^{-1}$ of $C_i$ ($x = 0$) for $D = 0.5 \times 10^{-10}$ m$^2$ s$^{-1}$, leaving about half of the origin solute flux at the crystal surface. This phenomenon is also evidenced by the nonuniform projection of precursor solution near the monocrystal (Supplementary Fig. 13). Interestingly, by increasing the concentration of the solute from 1 to 2 mol L$^{-1}$, the thickness of the diffusion layer is reduced by nearly half, which further promotes the diffusion of the monomer (Supplementary Fig. 14). Furthermore, this model was employed to investigate the effects of the substrate size, e.g., diffusion limit of the system, on mass transfer (Supplementary Fig. 15). The supply of solutes was considered to be unlimited or limited by setting different boundary conditions (see details in Supplementary Note 2). For a given growth process, the size effect is significant in small reaction containers with both boundary conditions, like those smaller than 1.5 mm. The concentration gradient and the length of the diffusion layer would gradually approach those obtained under infinite reaction boundary as the container size increased, and finally be the same at a certain container size. In our case, regardless of the monomer concentration on the boundary, the concentration distribution would barely be affected by the reaction limit when the substrate size is about three times larger than the final crystal size. These results emphasize the critical role of solute distribution and mass transfer in the growth process of thin monocrystal and present a potential guideline for monocrystal growth.

The growth process of perovskite monocrystals was recorded using an optical microscope (Fig. 2b). The side length of the HFG thin monocrystal increases linearly from 147–330 μm within 24 min, corresponding to the growth velocity of 8.0 μm min$^{-1}$, which exceeds that of the control thin monocrystal (Supplementary Fig. 16). The solute flux of HFG process is stable and achieves $2 \times 10^{-5}$ g mm$^{-2}$ min$^{-1}$, about 4 times of the control one (Supplementary Note 3). This trend also persisted on a larger scale, in which the surface area as a function of the growth time of the HFG monocrystal follows a quadratic relationship, with the length increasing from a few hundred microns to 2.3 mm (Supplementary Figs. 17 and 18). Furthermore, the captured movie data spanning the larger area verifies the fast growth velocity can be persistent for a long time in the HFG system (Supplementary Movie 1).

To further probe the mass transfer kinetics, the actual diffusion coefficient of monomers was estimated by UV–Vis absorption spectra, diffusion-ordered spectroscopy (DOSY), and dynamic light scattering (DLS) measurements. The schematic diagram of the local UV–vis spectroscopy experimental setup is shown in Fig. 2c, where the spatial distribution of lead halide species can be visualized by its characteristic peaks (Supplementary Fig. 19). In the HFG system, the rapid saturation of absorbance corresponds to a huge diffusion coefficient of $5.4 \times 10^{-10}$ m$^2$ s$^{-1}$, about triple of the control one (Supplementary Note 4). Subsequently, DOSY measurements were performed to determine the diffusion coefficient of the methylamine molecules in both two solution systems. The bigger diffusion coefficient was observed in the HFG system, up to $3.0 \times 10^{-9}$ m$^2$ s$^{-1}$, which is higher than $1.9 \times 10^{-9}$ m$^2$ s$^{-1}$ in the control system (Fig. 2d, Supplementary Note 5). Although the UV–Vis and NMR characterizations reflect the diffusion of different species, these results jointly confirm the fast diffusion of precursors in the HFG system in our work. Furthermore, identical trends can also be observed by DLS measurements and the parallel diffusion experiment from two space-confining substrates to pure solvents (Supplementary Note 6; Figs. 20 and 21).

We next explored the underlying reason for high mass diffusivity in glycol ether solvents, closely related to the colloidal and coordination chemistry. The ab initio molecular dynamics (AIMD) simulations of perovskite precursor species were performed in different solvents at 300 K (see detailed methods)[30]. The two model systems follow the same protocol: 3 lead iodide and 3 methylamine ions are evenly dispersed among 21 solvent molecules (Supplementary Fig. 22). At 8 ps, lead coordination polyhedrons in HFG solvent molecules are distributed as single molecules without agglomeration (Fig. 2e). By comparison, corner-sharing lead coordination polyhedron have formed in the control solvent model system at 4 ps. Then it undergoes further transformation into an edge-sharing lead coordination polyhedron. The non-aggregation property is a key factor in the formation of small colloidal clusters (Fig. 2f), which in turn leads to a large diffusion coefficient. To gain deeper insights into the electronic structure and local coordination environment of Pb species, X-ray absorption near edge structure (XANES) and extended X-ray absorption fine structure (EXAFS) analyses were conducted. The Pb L$_3$-edge XANES profiles suggest a similar oxidation state for Pb species in control and HFG systems (Supplementary Fig. 23). In the R-space Fourier-transformed EXAFS (Fig. 1g), both the control and HFG sample show two peaks around 2.15 Å and 2.67 Å, attributed to the scattering path of Pb-O and Pb-I bond, respectively[31]. Both bonds were found to be shortened in the HFG system, which again supports the results derived from the AIMD simulation (Supplementary Table 1 and Fig. 24). Concerning the higher donor number of 2-ME[32,33], this solvent is thought to better solvate perovskite precursors, and further contribute to smaller colloid.

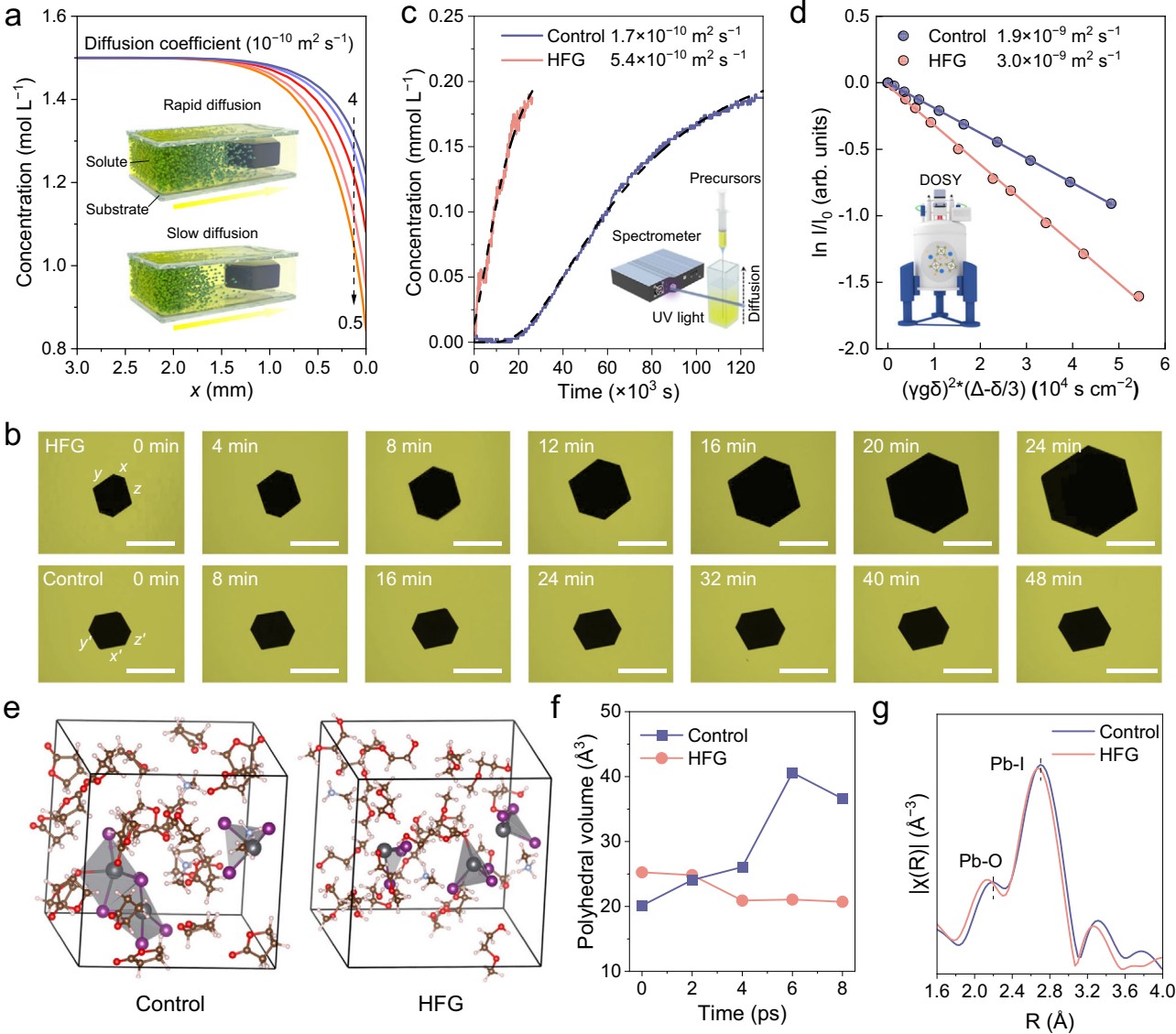

**Fig. 2 | Analysis of high-flux crystal growth mechanism. a** Simulated concentration field of solute near a MAPbI$_3$ monocrystal with varied diffusion coefficients ($0.5–4 \times 10^{10}$ m$^2$ s$^{-1}$). The inset shows the schematic illustrations of the growth process of thin monocrystal. **b** Optical images of the growth process of MAPbI$_3$ thin monocrystals from the control and high-flux growth (HFG). Scale bars: 300 μm. **c** Variation in time-dependent concentration of perovskite precursors that diffuse from the bottom of the cuvette. The inset shows the schematic diagram of the local UV–vis spectroscopy experimental setup. **d** The diffusion coefficient spectra of control and HFG system via NMR DOSY measurements. **e** Ab initio MD simulations of control and HFG model systems at 8 ps. The colors of the atoms are cyan: Pb, purple: I, brown: C, pink: H, light purple: N, red: O. All species are shown with ball and stick representations. **f** Calculated polyhedral volume of lead coordination polyhedrons in the HFG and control model systems. **g** Fourier-transformed spectra of Pb L$_3$-edge EXAFS spectra of control and HFG perovskite solution.

## The universality of the growth strategy

One of the most attractive features of the HFG method is the generality of various kinds of metal-halide perovskites, benefiting from their efficient solute diffusion and suitable coordination. A thin monocrystal library of metal-halide perovskites is shown in Fig. 3a with well-faceted geometries, including hexagons, rectangles, squares, and ribbons. Almost all commonly used cations, including MA, FA, Cs, 1-butylamine (BA), phenylethylamine (PEA), and butane-1,4-diaminium (BDA), and dimethylamine (DMA), can be applied to the thin monocrystal growth, which is inserted into the inorganic framework to form three dimensional, Ruddlesden−Popper, and Dion−Jacobson phases. Encouragingly, lead-free thin monocrystals can also be synthesized by replacing Pb with tin (II) (Sn), germanium (II) (Ge), antimony (III) (Sb), bismuth (III) (Bi), silver (Ag), and copper (Cu) with well-developed facets. Further growth of bromine (Br)- and chlorine (Cl)-based perovskites, such as Cs$_2$AgBiBr$_6$, PEA$_2$PbBr$_4$, FAPbBr$_3$, MAPbBr$_3$, and MAPbCl$_3$, requires the addition of some strong coordination solvents, e.g., N, N-dimethylformamide (DMF), or dimethyl sulfoxide (DMSO). Remarkably, perovskite alloys, such as MAPbI$_{2.8}$Br$_{0.2}$, FA$_{0.5}$MA$_{0.5}$PbI$_3$, Cs$_{0.05}$MA$_{0.95}$PbI$_3$, Cs$_{0.02}$FA$_{0.2}$MA$_{0.78}$PbI$_3$, and MAPbBr$_2$Cl, can be readily obtained, whose narrow PL emission reflects the structural homogeneity (Supplementary Fig. 25)[34]. One notable feature is that most monocrystals could grow to a millimeter scale, again demonstrating the feasibility of the HFG growth to different perovskites.

Interestingly, the growth of PEA$_2$PbBr$_4$ can be even observed by the naked eye, whose velocity attains 27.2 μm min$^{-1}$ at 40 °C (Supplementary Movie 2). In terms of monocrystal growth temperature, most monocrystals mentioned in this work can be obtained by a reasonable heating process below 70 °C (Fig. 3b; Supplementary Table 2), which avoids the large variations between the growth and application temperatures, and thus minimizes the internal thermal stress and defect density of monocrystals[35]. Only for some monocrystals with

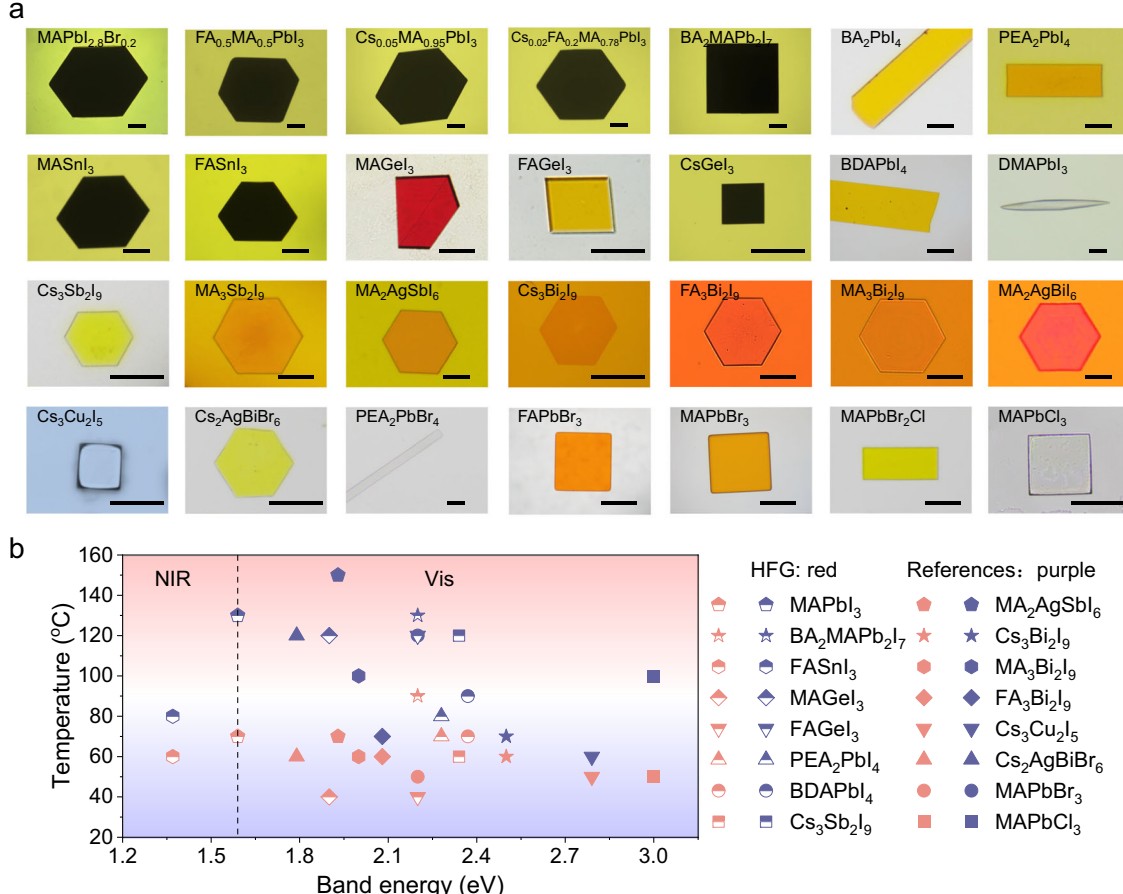

**Fig. 3 | Library of as-grown perovskite thin monocrystals via the high-flux approach. a** Optical images of 28 types of perovskite thin monocrystals from high-flux growth (HFG). Scale bars: 1 mm. **b** Summary of the synthesis temperature for the perovskite monocrystals.

thermodynamically unstable phases (such as $FA_{0.5}MA_{0.5}PbI_3$), a relatively high temperature (90 °C) is required to avoid the formation of non-perovskite phases. In addition, similar products can be harvested by using other glycol ether solvents (similar coordination state to 2-ME), such as 2-(2-methoxyethoxy) ethanol, triglycol monomethyl ether, and 2-(methylthio)-ethanol, which exhibits the larger diffusion coefficient and growth rate than GBL (Supplementary Figs. 26 and 27). Furthermore, the HFG monocrystals can be easily integrated on other rigid/flexible substrates, demonstrating the broad compatibility of our approach (Supplementary Fig. 28).

## Optoelectronic properties

We selected the triple-cation perovskite for optoelectronic applications because of its superior performance and phase stability[36]. The composition of the monocrystal is determined to be $Cs_{0.02}FA_{0.2}MA_{0.78}PbI_3$ through $^1$H NMR spectrum and inductively coupled plasma mass spectrometry (Supplementary Fig. 29). Mobility-lifetime ($\mu\tau$) product is one key figure-of-merit for X-ray and light detection, which can be obtained by fitting the photoconductivity with a modified Hecht equation[37]. As expected, a high $\mu\tau$ product of $2.53 \times 10^{-3}$ cm$^2$ V$^{-1}$ was attained for the HFG thin monocrystal device (Fig. 4a), about 6 times larger than the control one ($3.82 \times 10^{-4}$ cm$^2$ V$^{-1}$). The electrical resistivity of the HFG thin monocrystal is about $5.97 \times 10^9$ Ω cm (Supplementary Fig. 30), which ensures lower device noise levels. The Time-of-flight method was employed to reveal the hole and electron mobilities of thin monocrystal[38]. By fitting the transit time of the carrier at different bias voltages, the HFG thin monocrystal device attains hole mobility of 166.2 cm$^2$ V$^{-1}$ s$^{-1}$ and electron mobility of 153.4 cm$^2$ V$^{-1}$ s$^{-1}$, which is higher than the control one (Fig. 4b, c).

Combining the above data, the long hole diffusion length of the HFG thin monocrystal device (82.7 μm) exceeds its thickness (~50 μm), allowing efficient carrier collection without the assistance of applied bias.

The trap densities of as-grown thin monocrystals were then probed by space-charge-limited-current (SCLC) method[39]. The trap densities were evaluated to be $3.2 \times 10^9$ cm$^{-3}$ and $5.0 \times 10^{11}$ cm$^{-3}$ for the HFG and control devices, respectively (Fig. 4d). Thermal admittance spectra also illustrate that the trap density of states (tDOS) of the HFG thin monocrystal device is about one order of magnitude lower than the control one across the whole trap depth region (Fig. 4e)[40,41]. Steady-state photoluminescence measurements based on different incident directions and excitation wavelength further confirmed the trap reduction at both the surface and bulk of the HFG thin monocrystal (Supplementary Fig. 31). Therefore, we conclude that the efficient mass transfer can offer adequate precursor supply for crystal growth and suppress the generation of defects. The enhanced optoelectronic properties were also observed in the as-grown $MAPbI_3$ monocrystals (Supplementary Fig. 32), which reconfirms the significant role of mass transfer in crystal formation.

## Device performance

Given the superior optoelectronic properties of the HFG thin monocrystal, we further utilized it as an active layer to fabricate a self-driven thin monocrystal device with a p-i-n configuration of ITO/PTAA/perovskite/C60/BCP/Au for X-ray detection (Supplementary Fig. 33). A 60 μm thick monocrystal can totally attenuate the X-ray photons with a peak energy of 8 keV due to the high atomic number element enrichment of Pb (Fig. 5a). The device was then exposed to an X-ray

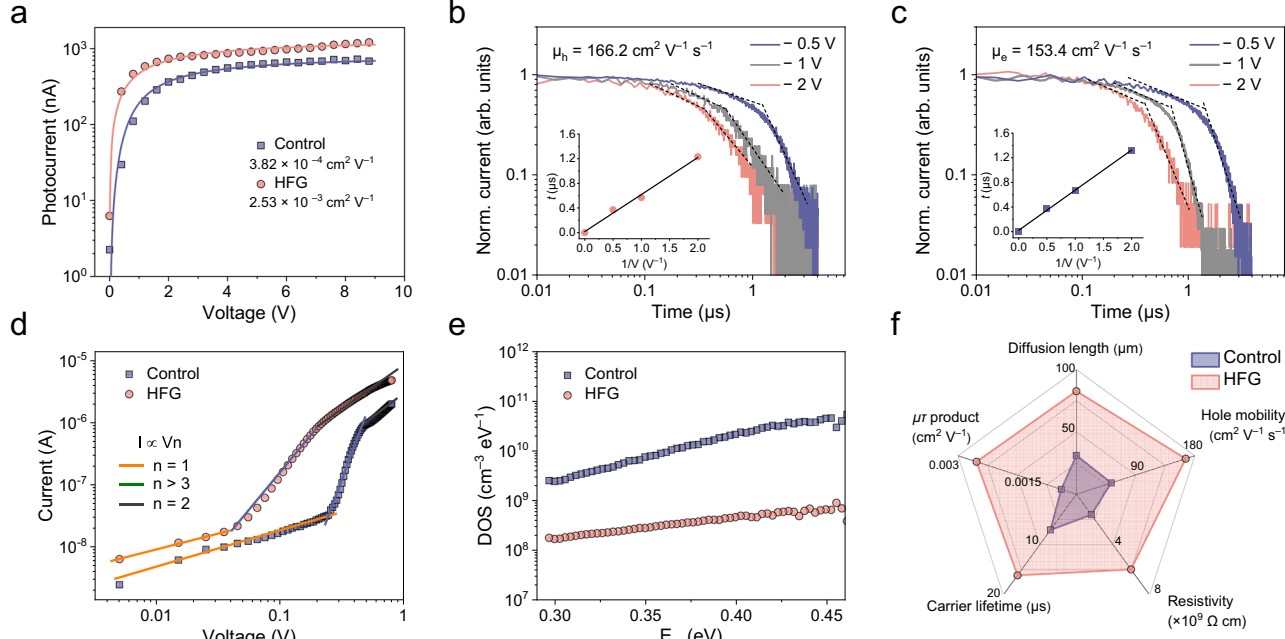

**Fig. 4 | Optoelectronic properties of thin monocrystals. a** Photoconductivity measurement of $Cs_{0.02}FA_{0.2}MA_{0.78}PbI_3$ thin monocrystal device from high-flux growth (HFG). Normalized time-of-flight **b** hole and **c** electron charge transient current curves of HFG thin monocrystal device under various reverse bias voltages. The inset shows the charge transit time versus the reciprocal of bias voltage. The excitation wavelength is 337 nm. **d** SCLC measurements of hole-only thin monocrystal devices. The device structure is ITO/PTAA/perovskite/spiro-OMeTAD/Ag. **e** Trap density of states of thin monocrystal devices. **f** Radar chart comparing diffusion length, hole mobility, resistivity, carrier lifetime, and $\mu\tau$ product for testing the optoelectronic properties of $Cs_{0.02}FA_{0.2}MA_{0.78}PbI_3$ thin monocrystals.

tube with a peak photon energy of 8 keV (Cu $K_\alpha$) and an operational voltage of 40 kV. Under X-ray irradiation, apparent current signals can be observed without any applied bias (Fig. 5b). The current density exhibits a linear correlation with the X-ray dose rate during irradiation (Fig. 5c). The sensitivity of this device achieved high values of $1.74 \times 10^5$ μC Gy$^{-1}$ cm$^{-2}$ under 0 V bias, and $1.01 \times 10^6$ μC Gy$^{-1}$ cm$^{-2}$ under 1.5 V bias, which is far more beyond than those of reported perovskite-based X-ray detectors (Fig. 5d and Supplementary Fig. 34). The sensitivity of HFG self-driven devices varied between $1.3 - 1.7 \times 10^5$ μC Gy$^{-1}$ cm$^{-2}$ under 0 V bias, indicating greater device-to-device reproducibility (Supplementary Fig. 35). Moreover, similar X-ray detection performance was also observed in MAPbI$_3$ monocrystal device (Supplementary Fig. 36). As another key parameter of X-ray detection, the lowest detection limit defined by international union of pure and applied chemistry (IUPAC), is the dose rate when the signal-to-noise ratio (SNR) = 3[42]. The self-driven device achieved the lowest detection limit of 11.8 nGy s$^{-1}$ (Fig. 5e and Supplementary Fig. 37), which is ~460 times less than the practical medical diagnosis[43].

Remarkably, the absence of baseline shift in almost all HFG thin monocrystal devices is particularly important (Fig. 5f), which has been usually observed in perovskite-based detectors[44]. We found that the high crystalline nature of thin monocrystals should significantly suppress the ionic conduction from both bulk (e.g., point defects, dislocations) and surface (e.g., grain boundaries)[11]. In addition, the response time of the device was much faster than the on/off speed of the X-ray source, owning to the indiscernible current decay. Negligible current variation was observed during long-term continuous operation under X-ray and 1.5 V bias electrolysis, revealing excellent radiation hardness. In contrast, the notable dark current shift can be seen for the control thin monocrystal. We also conducted an air stability study of the self-driven device without encapsulation under ambient atmosphere (RH = 25 ± 5 %). After 600 h, the response current of HFG thin monocrystal devices retained 95 % of the initial value (Supplementary Fig. 38).

As a proof-of-concept, the HFG thin monocrystal devices enable X-ray imaging under the zero-bias mode. The experimental setup is shown in Fig. 5g. The object is inserted between the X-ray source and the device, and then scanned along the X−Y direction to collect response current pixels. Under X-ray irradiation, metal pins embedded in the plastic can be observed due to the weak X-ray attenuation property of the plastic (Fig. 5i). Furthermore, nearly equal electrical signals were observed in a perovskite detector array with 6 × 6 pixels under dark and X-ray irradiation (Supplementary Fig. 39), validating the spatial uniformity of high-quality monocrystal. We further imaged a larger object, e.g., a 110 × 50 ×5 mm aluminum plate mask plate with a shark pattern. The obtained X-ray image exhibits excellent contrast (Fig. 5j), which is useful for many practical applications, e.g., medical imaging, security checks, and bioimaging[45,46].

## Discussion

In summary, our combined experimental and theoretical observations of perovskite solute diffusion provide direct evidence that mass transfer is the rate-determining step of space-confined crystal growth. We show that the solute diffusion velocity is closely related to the colloidal geometry of the metal-halide complex that can be understood by the Stokes-Einstein relationship. By tailoring the coordination chemistry using glycol ether solvents, we develop a high-flux growth technique that could successfully prepare 29 types of high-quality perovskite thin monocrystals at 40−90 °C with a growth velocity up to 27.2 μm min$^{-1}$ and lateral size of ~2 cm. The high solute flux also suppresses the bulk crystal defect formation caused by the mass transfer kinetics. As a result, the obtained $Cs_{0.02}FA_{0.2}MA_{0.78}PbI_3$ thin monocrystal presents an excellent monocrystalline nature with a low trap density of $2.5 \times 10^{10}$ cm$^{-3}$ and a large $\mu\tau$ product of $2.53 \times 10^{-3}$ cm$^2$ V$^{-1}$. The self-driven X-ray detector attains an impressive sensitivity of $1.74 \times 10^5$ μC Gy$^{-1}$ cm$^{-2}$ and the lowest detection limit of 11.8 nGy s$^{-1}$. This synthetic strategy sheds light on the fundamental growth mechanisms of hybrid perovskite monocrystals and provides

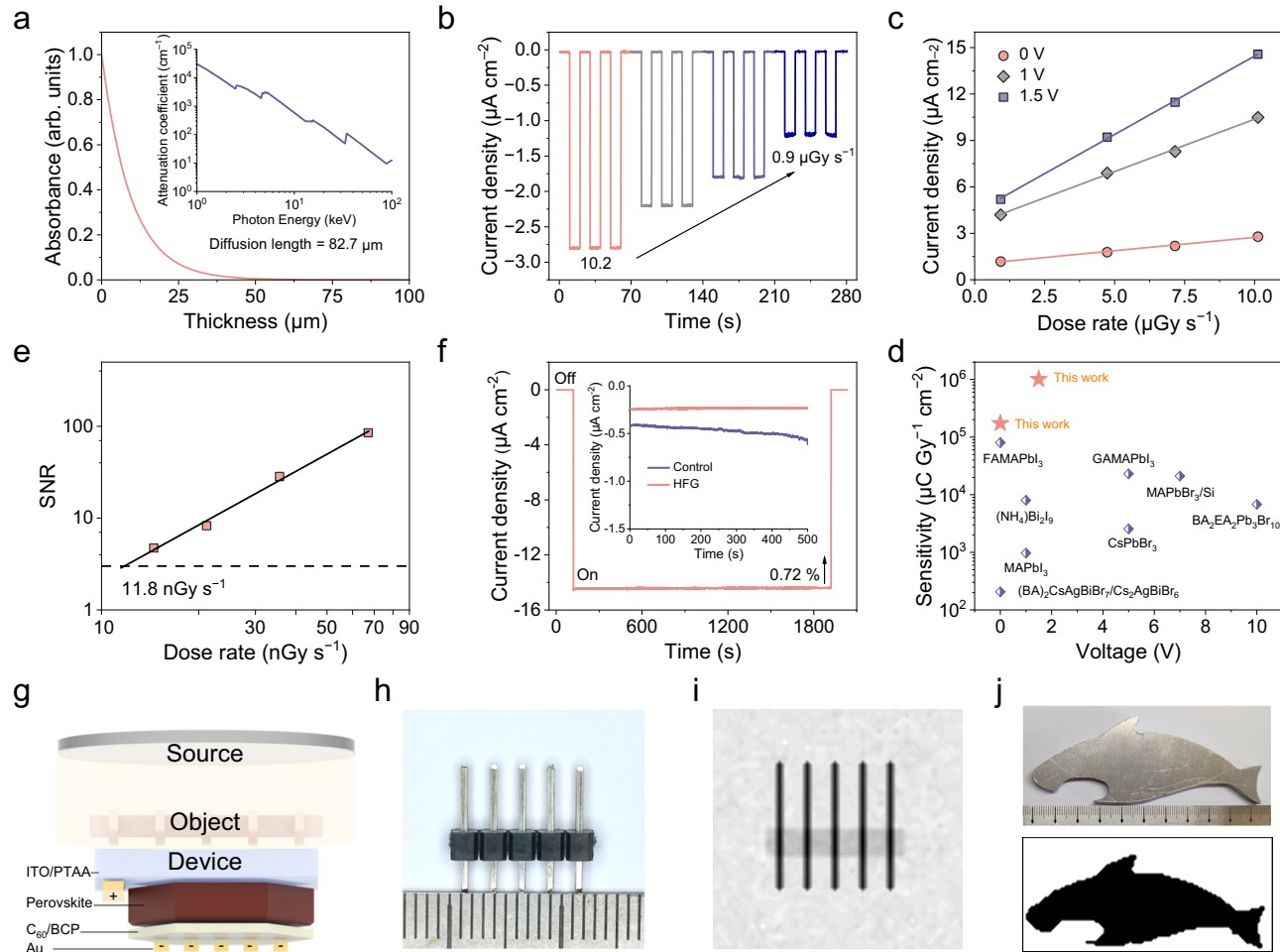

**Fig. 5 | X-ray detection performance and imaging of perovskite thin monocrystalline device. a** Attenuation coefficient of $Cs_{0.02}FA_{0.2}MA_{0.78}PbI_3$ to 8 keV X-ray photons. The inset shows the calculated attenuation efficiency. **b** Time-dependent response of the self-driven thin monocrystal device under various dose rates. **c** Output current density of thin monocrystal device from high-flux growth (HFG) under different dose rates. **d** Comparison of the X-ray response sensitivity of perovskite X-ray detectors. References are shown in Supplementary Tables 3 and 4. **e** SNR is dependent on the X-ray dose rate for the self-driven HFG thin monocrystal device. **f** Radiation stability measurement of the HFG thin monocrystal device under the X-ray dose rate of 10.2 µGy s$^{-1}$ and 1.5 V bias. The inset shows the dark current variation under 1.5 V bias. **g** Schematic illustration of X-ray/light projection imaging with the self-driven thin monocrystal device. **h** Photograph and **i** X-ray images of pin headers. The X-ray image was obtained by the self-driven thin monocrystal device under is 920 nGy s$^{-1}$. **j** Photograph and X-ray images of 5 mm thick aluminum plate mask plate with a dolphin pattern. The X-ray image was obtained by the self-driven thin monocrystal device under is 540 nGy s$^{-1}$.

promising thin monocrystalline perovskite materials for practical optoelectronic applications.

## Methods

### Materials

Lead (II) iodide (PbI$_2$, 99%), dimethyl sulfoxide (DMSO, 99.9%), poly[-bis(4-phenyl)(2,4,6-trimethylphenyl)amine] (PTAA), cesium iodide (CsI, 99.9%), tin (II) iodide (SnI$_2$, 99.99%), germanium (II) iodide (GeI$_2$, 99.9%), and silver iodide (AgI, 99.999%) were purchased from Sigma-Aldrich. 2-methoxyethanol (2-ME, 99%), N,N-dimethylformamide (DMF, 99%), lead (II) bromide (PbBr$_2$, 98%), lead (II) chloride (PbCl$_2$, 99%), cesium bromide (CsBr, 99%), bismuth (III) bromide (BiBr$_3$, 99%), silver bromide (AgBr, 99.9%), methylamine hydrochloride (MACl, 99%), and chlorobenzene (CB 99.5%) were purchased from Alfa Aesar. Bismuth (III) iodide (BiI$_3$, 98%), antimony (III) iodide (SbI$_3$, 99.999%), γ-butyrolactone (GBL, 99%), and copper (I) iodide (CuI, 98%) were purchased from J&K Scientific. Methylamine hydroiodide (MAI), formamidine hydroiodide (FAI), butylamine hydroiodide (BAI), dimethylamine hydroiodide (DMAI), phenethylammonium hydroiodide (PEAI), butane-1,4-diammonium iodide (BDAI$_2$), methylamine hydrobromide (MABr), formamidine hydrobromide (FABr), and

phenethylammonium hydrobromide (PEABr) were purchased from Xi'an Yuri Solar Co. LTD. 2-(methylthio)-ethanol (97%), 2-(2-methoxyethoxy) ethanol (98%), and triglycol monomethyl ether (97%) were purchased from Leyan. Ethanol (AR, 99.7%), acetone (99.5%), and toluene (99.8%) were purchased from Shanghai Titan Technology Co. LTD. Fullerene (C$_{60}$, 99.5%) was purchased from Nano-C. Bathocuproine (BCP, 99%) was purchased from Nichem.

### Monocrystal growth and device fabrication

Indium tin oxide (ITO) glass substrate, fluorine-doped tin oxide (FTO) glass substrate, glass substrate, mica substrate, and polyethylene terephthalate (PET) substrate were cleaned by ultrasonication in acetone, ethanol, and water for 20 min, respectively. Then, the substrates were treated with UV ozone for 15 min. PTAA was used for the hydrophobization of substrates. PTAA (2 mg L$^{-1}$ in toluene) was spin-coated onto substrates at 3000 rpm for 30 s, followed by being heated at 100 °C for 10 min. The growth of perovskite thin monocrystals is based on a low-temperature space-confined process. In a typical synthesis, the precursor solution for MAPbI$_3$ monocrystals was formulated by dissolving 0.922 g PbI$_2$ and 0.318 g MAI in 1 mL 2-ME, stirring at room temperature overnight. A hydrophobized substrate was coated with

precursor solution and then covered with another substrate. The precursor solution would be uniformly distributed due to the existence of surface tension. The substrates were heated by a hot plate with a ramp rate of 4 °C h$^{-1}$ from 50 °C to 70 °C. The growth process of 2-(methylthio)-ethanol, 2-(2-methoxyethoxy) ethanol, and triglycol monomethyl ether was similar to that of 2-ME. Unless otherwise stated, the growth process of the other thin monocrystals was similar to that of MAPbI$_3$ by using saturated solutions. The detailed synthetic condition can be seen in Supplementary Tables 5 and 6. The devices were fabricated by thermally evaporating C$_{60}$ (30 nm), BCP (6 nm), and Au (100 nm) on the back of a thin monocrystal. The effective area was determined by a metal mask.

## Solubility measurement

The solubility of MAPbI$_3$ in 2-ME was measured in the temperature range of 30–70 °C. Firstly, a saturated precursor solution was prepared by dissolving 31.1175 g PbI$_2$ and 10.7325 g MAI in 20 mL of 2-ME at room temperature. The filtered precursor solution (1 mL) was transferred to a closed vial by pipetting. The vial was kept in a preheated water bath for 5 h to ensure adequate precipitation of the solute. Then, the supernatant (200 μL) was transferred to an open vial, which was placed on a 60 °C hot plate to evaporate the solvent in the N$_2$ atmosphere. The concentration is calculated from the supernatant volume and the mass of the solute. More than three samples were tested for each point to obtain accurate results. For super solubility measurement, we prepared MAPbI$_3$ precursor solution with different concentrations (e.g., 2.5 mol L$^{-1}$, 2.1 mol L$^{-1}$, 1.8 mol L$^{-1}$, 1.5 mol L$^{-1}$). The substrates were heated from 30 to 70 °C at a rate of 4 °C h$^{-1}$, and the crystallization temperature was recorded to estimate the super solubility curve.

## Diffusion coefficient measurement

For diffusion measurement, the in-situ absorption spectroscopy equipment was built, which consists of a spectrometer (Ocean Optics, QE Pro), a cuvette holder (Ocean Optics, CUV-ALL), optical fibers, an optical fibers attenuator (FVA-UV), and a light source (Deuterium and tungsten halogen, DH2000-LL). The perovskite precursor solutions were diluted to 0.03 mol L$^{-1}$. The precursor solution, pure solvent, cuvette, and cuvette holder were placed on a hot plate, which was preheated at crystal growth temperature for 1 h. Before injecting the cuvette, we need to first immerse the pipette tip in another pure solvent to eliminate the contribution to the absorbance of the liquid contaminated by the tip of the pipette. The pure solvent (3.2 mL) was transferred to the cuvette to collect the reference spectrum. Then, the precursor solution (20 μL) was carefully injected into the bottom of the cuvette. Due to the existence of the concentration gradient, the perovskite precursor will gradually diffuse to the upper layer of the cuvette and finally achieve a uniform distribution. In addition, we took the substrates sandwiched with the precursor solution into the pure solvent, which is preheated at its crystal growth temperature. To examine the diffusion of perovskite precursor, the color change of the substrates and the solution was recorded using a digital camera (Sony IMX586). The hydrodynamic particle size of perovskite precursor solutions was measured by dynamic light scattering (DLS) using a Malvern Zetasizer Nonoseries (Nano ZS90). All NMR spectra were collected in a Bruker 500 MHz AV NMR. 500 μL DMSO D$_6$ with dissolved triple perovskite monocrystal was transferred from the reaction system to the nuclear magnetic tube and collected at 276 K to obtain the $^1$H NMR spectra (dummy scans = 0, number of scans = 32, receiver gain = 4). For the NMR diffusion-order spectroscopy (DOSY) experiments, perovskite solutions contained acetonitrile-D$_3$ (500 μL, vol% 9:1) in the nuclear magnetic tube. DOSY spectra (dummy scans = 4, number of scans = 8, receiver gain = 4) of HFG and control solutions were collected at room temperature and 70 °C, respectively. The data post-processing is completed using the Dynamics Center 2.8.3 and Topspin 3.6.1.

**Computational calculations**. First-principles calculations based on density functional theory (DFT) were conducted using the Vienna ab initio simulation package (VASP). The projector-augmented wave (PAW) method is used to depict the interaction between the valence electrons and the ionic cores. The Perdew–Burke–Ernzerhof (PBE) functional approximated the exchange-correlation energy in the framework of the generalized gradient approximation. Ab initio molecular dynamics (AIMD) simulations were performed with a plane-wave cutoff energy of 300 eV, for which the canonical ensemble is used. The canonical ensemble is a collection of systems characterized by the same values of a number of particles, volume, and temperature. The integration time step of the dynamics was set to 1 fs. AIMD runs of at 300 K were used in a cube supercell containing three MAPbI$_3$ and twenty-one solvent molecules. The supercell sizes of 2-ME and GBL solvent system are 17.9 × 17.9 × 17.9 Å and 15.9 × 15.9 × 15.9 Å, respectively.

**Structure characterization**. The morphology of perovskite thin monocrystals was assessed through field emission scanning electron microscopy (FESEM, HITACHI S4800) and atomic force microscope (AFM, NT-MDT). Electron backscatter diffraction (EBSD) patterns were measured using the EBSD detector of Oxford Instruments on a NOVA NanoSEM 230 at 20 kV with the sample tilted at 70°. The AZtec software was used to analyze crystal orientation. The optical images of perovskite thin monocrystals were recorded by an upright fluorescence microscope (LW450LFT-LED) with a Sony digital fluorescent camera. The thin monocrystals were photographed in projection mode, excluding those grown on the silicon wafer. Transmission electron microscopy (TEM) characterization was conducted using Thermo Fisher Talos F200X microscope under 200 kV. High angle annular dark field (HAADF)-STEM images were performed on a convergence semi-angle of 11 mrad, and inner- and outer collection angles of 59 and 200 mrad, respectively. Energy dispersive X-ray spectroscopy (EDX) was measured using four in-column Super-X detectors. The preparation process of TEM samples was performed in a nitrogen glove box. The thin monocrystals were scraped from the substrate, ground into powder, and finally dispersed them into toluene. XAFS data including XANES and EXAFS at Pb L$_3$-edge were measured at room temperature on the 1W1B beamline at the Beijing Synchrotron Radiation Facility (BSRF). The crystallographic information was investigated by powder X-ray diffraction (Bruker Advance D8 X-ray diffractometer, Cu Kα radiation, 40 kV, 40 mA). X-ray diffraction rocking curve, phi scan curve, and pole figures measurements were performed by using Bruker D8 Discover diffractometer with centric eulerian cradle (Cu Kα radiation, 3 kW, lynxeye array detector). A pole figure consists of a series of phi scan dates (0–360°, 5° resolution) with fixed 2θ and χ. In our work, 2θ is fixed at 40.7°, χ varied in steps of 5° in the range of 0 to 85° to ensure coverage over the entire pole sphere. Fourier-transform infrared spectra (FT-IR) were collected from Thermo Nicolet 6700. For contact-angle measurement, 5 μL water was carefully dropped on the hydrophobized substrate by the Dataphysics OCA20 contact-angle system. The absorption spectra of perovskite precursor solutions were measured by using a Cary 500 UV–Vis–NIR spectrophotometer, in which precursor solutions were diluted to 1/8000 of the crystal growth concentration.

## Photoelectronic characterization

Steady photoluminescence (PL) spectra of perovskite thin monocrystals were investigated by using the QE Pro spectrometer (Ocean Optics) with an excitation wavelength of 365 nm and 635 nm. Transient photovoltage (TPV) decay curves of the thin monocrystal devices were recorded by using 1 GHz digital oscilloscope (Agilent DSO-X 3104 A) under AM 1.5 G irradiation (100 mW cm$^{-2}$) and 4 ns width, 337 nm laser pulses (SRS N$_2$ laser). Capacitance-frequency characteristics of the thin monocrystal devices were measured by an LCR meter (Agilent,

E4980A). The time of flight (ToF) method was employed to measure the carrier mobilities of thin monocrystal devices with the structure of ITO/PTAA/perovskite/$C_{60}$/BCP/Au (20 nm). In the dark, the device was illuminated by a 337 nm laser pulse (SRS $N_2$ laser) under different bias voltages. The current signal was recorded by using a low-noise current preamplifier (SR570) and 1 GHz digital oscilloscope (Agilent DSO-X 3104 A). By changing the incident direction of the laser, we can obtain hole and electron mobilities, respectively. For μτ product measurement, a 635 nm LED as excitation light, is modulated at 50 Hz by a function generator. The photoconductivity current of a thin monocrystal device was recorded by using a Keithley 2400 digital source meter.

### Detector performance measurements
During the X-ray detection experiment, the thin monocrystal devices were exposed to a Cu X-ray tube (Canon, A40) with a tube voltage of 40 kV. The dose rates were controlled by adjusting the tube current (2–40 mA) and aluminum foil, which was calibrated by RaySafe X2 R/F sensor. The X-ray response current was measured using a Keithley 2400 digital source meter.

### Imaging measurements
For X-ray imaging measurement, the pin header is inserted between the X-ray tube and the self-driven thin monocrystal device. The object was moved pixel by pixel through an X–Y linear stage (RXSN40-100). The current signal of the device was recorded by a Keithley 2400 digital source meter.

## Data availability
The data within Supplementary Information are available from the corresponding authors. Source data are provided in this paper.

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

## Acknowledgements

This work was financially supported by National Ten Thousand Talent Program for Young Top-notch Talent, National Natural Science Fund for Excellent Young Scholars (52022030), National Natural Science Foundation of China (51972111, 22379044, 52203330, 12304109, 22274052), Shanghai Pilot Program for Basic Research (22TQ1400100-5), "Dawn" Program of Shanghai Education Commission (22SG28), Shanghai Municipal Natural Science Foundation (22ZR1418000), the Science and Technology Innovation Plan of Shanghai Science and Technology Commission (), Shanghai Sailing Program (22YF1413100, 22YF1410000), Postdoctoral Research Foundation of China (2021M701190), the Fundamental Research Funds for the Central Universities (JKD01231632, JKVD1231041), and Shanghai Engineering Research Center of Hierarchical Nanomaterials (18DZ2252400). S.D. acknowledges the support by the Shanghai Rising-star Program (20QA1402400) and the Program for Professor of Special Appointment (Eastern Scholar) at Shanghai Institutions of Higher Learning. Additional support was provided by the Feringa Nobel Prize Scientist Joint Research Center and Shanghai Frontiers Science Center of Optogenetic Techniques for Cell Metabolism. The authors also thank the crew of the 1W1B beamline of the Beijing Synchrotron Radiation Facility (BSRF) for their constructive assistance with the XAFS measurements and data analyses.

## Author contributions

Y.H. and S.Y. conceived the research. D.L. designed and conducted the experiments. Y.Z. carried out the theoretical simulation. X.Y.S. and H.Y. performed the theoretical calculations. X.F.W. and S.D. contributed to the TEM characterization. C.Z. helped with device performance measurement. S.Y., X.L. and M.L. assisted in the Figure preparation. D.L. and Z.P.W. performed the structure characterization. H.Z. and Y.Y contributed to the DOSY measurement. Y.H., S.Y. and D.L. wrote the manuscript. Y.P. and C.Z. assisted with manuscript revision. Y.H., S.Y. and H.G.Y. provided all the support needed in this work. All authors contributed to the general discussion and reviewed the paper.

## Competing interests

The authors declare no competing interests.
