## [Peer Review File · Nature Communications]

Universal growth of perovskite thin monocrystals from high solute flux for sensitive self-driven X-ray detectionREVIEWER COMMENTS

Reviewer #1 (Remarks to the Author):

The manuscript reveals that mass transfer is one major limiting factor during solution growth of perovskite thin monocrystals. In brief, the author designs a growth strategy based on high solute flux to overcome this limitation, enabling the universal growth of a library of perovskite thin monocrystals via a low-temperature ($<90^{\circ}\text{C}$), rapid (up to $27.2\ \mu\text{m}/\text{min}$) manner. The fast mass transfer guarantees the uniform supply of precursors, and suppress the bulk defect formation of perovskite monocrystals. And the as-grown perovskite monocrystal deliver a record X-ray sensitivity of $1.74 \times 10^5\ \mu\text{C Gy}^{-1}\ \text{cm}^{-2}$ without applied bias. This synthetic strategy provides promising thin monocrystalline perovskite materials towards the practical opto-electronic applications. The following comments are suggested to be addressed.

1. In this manuscript, the authors reveal that mass transfer limits the overall formation of perovskite thin monocrystals. However, the authors demonstrated the claim only by comparing two solvent systems including 2-methoxyethanol (high solute flux, HFG) and γ -butyrolactone (control sample). The authors indicate that there are other glycol ethers solvents can achieve the preparation of thin monocrystals (Supplementary Fig. 18). The author should provide more experimental results of other solvent systems to fully demonstrate this mass transfer mechanism.
2. In Supplementary Fig. 18, the authors present the optical microscope images of MAPbI₃ thin monocrystal grown from glycol ether solvents. However, from the results, the monocrystals grown by other glycol ether system are not as large as the control system. The authors should provide an explanation.
3. Supplementary Fig. 5 shows that the crystal thickness can be controlled in a range of 4 – 50 μm by applying loads. As we know, thinner single crystals ($\leq 1\ \mu\text{m}$) have a wider range of application scenarios, such as Solar cells, Photodetectors and LEDs. Is it possible to prepare thinner single crystal film by controlling the pressure? Will the reduction of single crystal thickness affect the crystallization quality of the final thin monocrystal?
4. The authors provide the photographic screenshot of a single thin monocrystal. In-situ microscopic characterization of the growth of thin monocrystals over a wide range is required. Additionally, the image of the thin monocrystals on the entire substrate should be given in SI besides the selected crystal.
5. Fig.2a shows the schematic illustrations of the growth process of thin monocrystal. However, I think this figure is of little significance for understanding the mechanism of mass transfer, so it is suggested to draw the mechanism diagram together with the related experimental data. The growth mechanism of the Control and HFG thin monocrystals should be compared.
6. Figure 1 provides the structural characterizations of the HFG thin monocrystals. For comparison, the structural characterization (such as XRD, TEM and surface roughness) of the Control thin monocrystals should be provided in SI.
7. The interface is an important factor affecting the quality of the final crystal in the process of growing single crystal by space-confined method. The authors should provide the roughness and wettability of the substrate.
8. In Fig. 1g, the authors indicate that the thickness of the thin monocrystal is about 50 μm , corresponding to a record-high aspect-ratio of 379 for MAPbI₃ perovskite. However, the corresponding supporting materials are not provided. The authors should provide aspect-ratio data reported in other work.
9. In this work, the authors mention that the solubility of MAPbI₃ in 2-methoxyethanol is 2.5 M, and the author used a solution concentration of 1-2 M to grow thin monocrystals. Why did

the authors not choose a concentration greater than 2 M to grow monocrystals?

10. The author should provide the electron backscatter diffraction (EBSD) pattern with crystal boundaries (corresponding to Fig. 1f) in SI.

11. The author ought to provide a cross-section SEM images of the self-driven thin monocrystal device.

12. The thickness of the thin monocrystals (4-50 μm) in this work does not fulfill the requirements for TEM testing. The authors are requested to add the sample preparation process for TEM testing in the Structure characterization section.

13. In order to increase the standardization of the manuscript, it is recommended that the units be revised to a uniform format, such as mol/L should be revised to mol L⁻¹.

14. In this work the authors used the prepared monocrystal to record X-ray imaging. As we know, X-ray imaging for medical imaging, security check, and bioimaging, usually need large uniform film. Is it possible to present X-ray imaging over large area using the prepared monocrystal?

Reviewer #3 (Remarks to the Author):

The paper presents a modified approach to developing perovskite monocrystals through a solvent composition modification method that enables improved growth rates, supports lower defect levels and can be adopted to a variety of compositions. The work itself builds on considerable foundations in monocrystal growth, which includes solvent and composition tunability. Through high level optoelectronic characterisation, it is shown that the synthetic approach yields significant improvements in mobility, lifetime and $\mu\tau$ product. This is capitalised through the fabrication of X-ray detectors in a vertical diode configuration and achieve very high sensitivity factors and low detection limits.

The work itself is interesting and presented at a high level. However, it bridges several scientific fronts with limited details in the key areas.

Firstly, the diffusion model presented is highly simplistic in its consideration and the discussion section around it is equally so. The fundamental nature of the precursors is suggested based on DLS, which has questionable validity at the sub nanometer level. No examination of the actual speciation has been conducted from a chemistry perspective, making this entire section highly speculative. I would think that comparative NMR would for instance provide an ability to determine diffusion in a much more accurate way. The consideration between the interplay of diffusion and reaction limits is not well presented or considered, and certainly not compared between precursors.

The use of the so called high flux growth must be highly dependent on the interplay between the solvent and the precursor chemistry. However, this is not explored, nor are any of the comparative growth considerations across the compositions presented. What is shown, namely the bandgaps vs temperature and optical images, should be expanded to provide depth to the comparison, for instance, growth rates, stability factors, structure etc. Through this comparison, one can make a more generalised growth mechanism. With the data presented, there is insufficient evidence to validate this, merely to show that the approach works.

The work on the optoelectronic properties is well conducted and presented. Minor points around this: details of the exact crystal sizes nor how the composition was determined aren't clearly presented.

The x-ray detector data suggests impressive device performance values. However, the devices weren't actually shown, and it would be useful to better understand what the effective area was of the devices and how they were actually fabricated. For instance, you used a shadow mask to deposit onto a 400-500 micron size pixel? How did you avoid shorting? Statistic data is not show here, and it would be useful to get a broader comparison to ensure that there aren't measurements errors.

General remarks:

It's not entirely clear what error margins are on the diffusion experiment presented in Fig 2. Further details around this should be included, particularly around the reproducibility and the impacts of the precursor addition methodology.

In general, the experimental errors are not presented across the paper.

The light based detector approach lacks any impact from my perspective and feel like a tack on. It would be more favourable to focus on the X-ray detector enhancements and statistics, rather than broader yet again into another section without completion of the others.

Reviewer #1 (Remarks to the Author):

Comment 1: The manuscript reveals that mass transfer is one major limiting factor during solution growth of perovskite thin monocrystals. In brief, the author designs a growth strategy based on high solute flux to overcome this limitation, enabling the universal growth of a library of perovskite thin monocrystals via a low-temperature (<90°C), rapid (up to 27.2 $\mu\text{m}/\text{min}$) manner. The fast mass transfer guarantees the uniform supply of precursors, and suppress the bulk defect formation of perovskite monocrystals. And the as-grown perovskite monocrystal deliver a record X-ray sensitivity of $1.74 \times 10^5 \mu\text{C Gy}^{-1} \text{cm}^{-2}$ without applied bias. This synthetic strategy provides promising thin monocrystalline perovskite materials towards the practical opto-electronic applications. The following comments are suggested to be addressed.

1. In this manuscript, the authors reveal that mass transfer limits the overall formation of perovskite thin monocrystals. However, the authors demonstrated the claim only by comparing two solvent systems including 2-methoxyethanol (high solute flux, HFG) and γ -butyrolactone (control sample). The authors indicate that there are other glycol ethers solvents can achieve the preparation of thin monocrystals (Supplementary Fig. 18). The author should provide more experimental results of other solvent systems to fully demonstrate this mass transfer mechanism.

Response: We appreciate the reviewer's insightful suggestions and agree that it would be of importance to clarify the mass transfer in other glycol ethers. We firstly recorded the UV-vis absorption spectra of the perovskite precursor to elucidate the coordination state in a glycol ether solvent, including 2-(methylthio)-ethanol (abbreviated as 2-MSE), 2-(2-methoxyethoxy) ethanol (abbreviated as 2-2ME), and triglycol monomethyl ether (abbreviated as 3-ME). As shown in Fig. R1, all glycol ether solvents exhibit similar lead halide characteristic peaks, specifically PbI_2 and PbI^{3-} , indicating the similar coordinate mode with all these glycol ethers. Simultaneously, the colloid size also plays a pivotal role in the mass transfer process during crystal growth. In comparison to 2-ME, other glycol ether solvents displayed slightly increased colloid sizes between 0.83–0.96 nm (Fig. R2). It is noteworthy that these values still remain to be lower than the colloid size observed in GBL system of ~ 1.5 nm.

Fig. R1 | UV-vis absorption spectra of the MAPbI₃ dissolved in different glycol ether solvents.

Fig. R2 | Colloidal hydrodynamic size distribution via dynamic light scattering of perovskite precursor solutions. Perovskite precursor in these glycol ether solvents exhibits small micelle diameters as well.

Based on the aforementioned results, we systematically assessed the diffusion coefficient of the glycol ether solvent at the crystal growth temperature (70°C). To

enable a direct comparison of diffusion coefficients across different solvent systems, we utilized specific volume of solutions (10 μL) with identical concentrations (0.03 mol L^{-1}) and injected them into the respective pure solvents (2 mL). We then monitored the changes of absorbance value at characteristic wavelengths to obtain the diffusion coefficient of the glycol ether solvent, which falls between that of 2-ME and GBL systems. As shown in Fig. R3, the diffusion coefficients of 2-MSE and 2-ME are approximately equal, while those of 2-2ME and 3-ME stand at $3.9 \times 10^{-10} \text{ m}^2 \text{ s}^{-1}$ and $2.3 \times 10^{-10} \text{ m}^2 \text{ s}^{-1}$, respectively. We attribute this variation to the increase in chain length and solvent viscosity^{1,2}, which results in the change in colloidal size and the reduction in mass transfer, yet these values remain higher than the diffusion coefficient of the GBL system at 130°C.

Fig. R3 | Variation in time-dependent concentration of perovskite precursors that diffuse from the bottom of cuvette.

Comment 2: 2. In Supplementary Fig. 18, the authors present the optical microscope images of MAPbI_3 thin monocrystal grown from glycol ether solvents. However, from the results, the monocrystals grown by other glycol ether system are not as large as the control system. The authors should provide an explanation.

Response: We thank the reviewer for the comments. We have just tried to grow perovskite monocrystals using these solvents with very short time intervals of less than 12 hours. This is the reason why the monocrystals present in Supplementary Fig. 18 is only in the hundreds of micron range. By extending crystal growth time, we successfully achieved monocrystals with sizes exceeding 1 mm (Fig. R4). Additionally,

we have supplemented the study with a detailed investigation of the mass transfer process as suggested by the reviewer in *Comment 1*.

Fig. R4 | Molecular structure of glycol ether solvents similar to that of 2-methoxyethanol. Optical microscope images of MAPbI₃ thin monocrystal grown from glycol ether solvents for 24 h. Scale bars: 300 μ m.

Comment 3: 3. Supplementary Fig. 5 shows that the crystal thickness can be controlled in a range of 4 – 50 μ m by applying loads. As we know, thinner single crystals (≤ 1 μ m) have a wider range of application scenarios, such as Solar cells, Photodetectors and LEDs. Is it possible to prepare thinner single crystal film by controlling the pressure? Will the reduction of single crystal thickness affect the crystallization quality of the final thin monocrystal?

Response: We greatly appreciate the constructive suggestions from the reviewer. For photodetector devices, the optimal thickness of the active layer should slightly exceed the light absorption depth, which determines the minimum thickness for harvesting an adequate amount of light^{3,4}. Therefore, reducing the thickness of the active layer can minimize the recombination of photo-generated charge carriers, leading to enhanced device performance. As mentioned in our manuscript, we achieved approximately 4 μ m thick monocrystal by applying loads. Moreover, to further decrease the thickness of monocrystal, we continued to increase the pressure. We obtained the monocrystal with the thickness of approximately 1 μ m, and the correlation between the pressure and thickness is summarized in the Fig. R5. X-ray diffraction (XRD) spectrum further verifies the crystallinity of the thinned monocrystals in Fig. R6, which shows the strong X-ray diffraction intensity of 60000, indicating that reducing the thickness of the single crystal does not adversely affect its crystal quality.

Fig. R5 | **a**, Cross-sectional SEM images of the HFG MAPbI₃ thin monocrystals under different pressure. **b**, Thickness of thin monocrystals as a function of pressure.

Fig. R6 | XRD spectrum of the 1 μm thick MAPbI₃ thin monocrystal prepared by HFG method.

Comment 4: 4. The authors provide the photographic screenshot of a single thin monocrystal. In-situ microscopic characterization of the growth of thin monocrystals over a wide range is required. Additionally, the image of the thin monocrystals on the entire substrate should be given in SI besides the selected crystal.

Response: Following the reviewer's suggestion, we captured the videos of HFG monocrystal growth, along with select snapshots featured in Fig. R7. The surface area

of the HFG monocrystal gradually increased at 70 °C, with the maximum size increasing from a few hundred microns to 2.3 mm after 90 min of growth time. We recorded the surface area of HFG monocrystal at different time points and plotted it in Fig. R8, which follows a quadratic relationship. This result can be attributed to the gradual increasing of the side areas and the stable mass transfer per unit the side area. Moreover, we provided the image of the monocrystals on the entire substrate. (Fig. R9).

Fig. R7 | Optical images of the growth process of the HFG MAPbI₃ thin monocrystals. Scale bar is 1 mm.

Fig. R8 | The area as a function of the growth time for the HFG MAPbI₃ thin monocrystal. The area of HFG monocrystal can be fitted by a quadratic function, indicating a stable solute flux.

Fig. R9 | Optical image of the monocrystals on the entire substrate.

Comment 5: 5. Fig.2a shows the schematic illustrations of the growth process of thin monocrystal. However, I think this figure is of little significance for understanding the mechanism of mass transfer, so it is suggested to draw the mechanism diagram together with the related experimental data. The growth mechanism of the Control and HFG thin monocrystals should be compared.

Response: We are thankful for the reviewers' careful examination of our figures. As suggested by the reviewer, we placed the schematic illustration as an inset in the simulation data and added a comparison of the two systems (Fig. R10). This schematic illustration depicts the correlation between crystal growth and diffusion as given in the manuscript. Specifically, the large diffusion coefficient and solute concentration are favorable for the mass transfer according simulation data and offer high solute flux available for the subsequent monocrystal growth. In other words, the accessibility of monomers at the crystal surface determines the crystal growth velocity.

Fig. R10 | Simulated concentration field of solute near a MAPbI₃ monocrystal with varied diffusion coefficients ($0.5\text{--}4\times 10^{10}\text{ m}^2\text{ s}^{-1}$). The inset shows the schematic illustration of the growth process of thin monocrystal for rapid and slow diffusion.

Comment 6: 6. Figure 1 provides the structural characterizations of the HFG thin monocrystals. For comparison, the structural characterization (such as XRD, TEM and surface roughness) of the Control thin monocrystals should be provided in SI.

Response: We agree with the reviewer that the structural characterization of the control monocrystals was incomplete, and we have supplemented them accordingly, as suggested by the reviewers. The thickness of the control one is identical to the HFG one of about 50 μm . Firstly, the XRD spectra show that no significant diffraction peak shift is found in the HFG crystals compared to the control crystals, but their diffraction intensity is enhanced almost twofold (Fig. R11). Secondly, we also performed high-resolution transmission electron microscopy high-resolution. As shown in Fig. R12, the control sample also exhibit the identical d -spacing of 0.38 nm, indicating that the difference in solvent type does not affect the nature of the perovskite crystal.

Fig. R11 | XRD patterns of the control and HFG thin monocrystals.

Fig. R12 | HRTEM image and the corresponding fast Fourier transform (FFT) pattern of the control thin monocrystal.

Fig. R13 | AFM height image of (a) HFG and (b) control thin monocrystals.

The surface quality of the thin monocrystal was evaluated using the atomic force microscope (AFM). The HFG crystal demonstrate an ultra-smooth surface with the root-mean-square roughness of 1.12 nm (Fig. R13), which is much lower than that of the control crystal sample (4.42 nm). This phenomenon can be attributed to two factors. Firstly, the control system has an insufficient supply of perovskite monomers, resulting in a high density of crystal defects. Secondly, the persistent and residual solvates located in both bulk and interface of the monocrystal due to the use of GBL solvent. This causes the formation of micron-sized holes during the solvent escape process, resulting in poor surface quality^{5,6}.

Comment 7: 7. The interface is an important factor affecting the quality of the final crystal in the process of growing single crystal by space-confined method. The authors should provide the roughness and wettability of the substrate.

Response: We would like to thank the reviewer's comments. It has been researched how substrate wettability impacts the formation of thin single crystals. Chen *et al.* conducted a comparative study on ion diffusion rates between hydrophobic and hydrophilic substrates, and observed that the hydrophobic substrates not only enhanced precursor diffusion but also reduced nucleation density^{7,8}. In this work, we used hydrophobic PTAA coated substrate, which has a wetting angle of up to 101.5° for water (Fig. R14). According to Young-dupré's equation⁹, the larger wetting angle corresponds to lower adhesion between the solution and substrate, facilitating the diffusion of the perovskite precursor solution in the micron-scale gap. Furthermore, the PTAA-covered ITO substrate shows an ultra-flat surface with the root-mean-square roughness of 1.61 nm, guaranteeing the flatness of the perovskite thin monocrystals (Fig. R15).

Fig. R14 | Contact angle measurements of the water on PTAA-covered ITO substrate.

Fig. R15 | AFM height image of the PTAA-covered ITO substrate.

Comment 8: 8. In Fig. 1g, the authors indicate that the thickness of the thin monocrystal is about 50 μm , corresponding to a record-high aspect-ratio of 379 for MAPbI_3 perovskite. However, the corresponding supporting materials are not provided. The authors should provide aspect-ratio data reported in other work.

Response: We thank the reviewer for the question. As shown in Table R1, we conducted a detailed comparison of aspect ratio data for MAPbI_3 thin monocrystals grown by various methods, including the lithography-assisted epitaxial-growth, and space confined growth method. The lithography-assisted epitaxial-growth method enable the fabrication of large-area quasi-monocrystals (high aspect ratio of up to 2750)¹⁰, but they rely on complicated lithography, huge bulk monocrystal substrates and may consist many dislocations and grain boundaries. In comparison, space-confined growth method can obtain the thickness-controllable single crystal wafers and achieve the direct integration between single crystal and conductive substrate. In this work, we introduced

a high-flux growth method to obtain a perovskite single crystal with a thickness of 50 μm , a length reaching 2 cm, and an impressive aspect-ratio exceeding 379. This finding surpasses other reported MAPbI₃ thin monocrystal materials^{11–15}. Song *et al.* introduced a low surface energy perfluorinated-gel modification layer as the soft contact layer on the growth substrates, and obtain monocrystal with a thickness of several hundred micrometers and a length of 4 cm, with a calculated aspect ratio of less than 400¹⁶.

Table R1 | Summary of the aspect ratio of MAPbI₃ monocrystal in recent reports.

No.	Year	Method	Later size (μm)	Thickness (μm)	Aspect-ratio	Ref.
1	2020	Lithography-assisted epitaxial-growth	55000	20	2750	[10]
2	2016	space-confined growth method	5840	150	39	[11]
3	2017	space-confined growth method	~2865	11.7	245	[12]
4	2018	space-confined growth method	544	2.5	218	[13]
5	2022	space-confined growth method	5000	200	25	[14]
6	2023	space-confined growth method	~4174	40	104	[15]
7	2023	space-confined growth method	40000	>100	<400	[16]
8	/	Space confined	18950	50	379	This work

Comment 9: 9. 9. In this work, the authors mention that the solubility of MAPbI₃ in 2-methoxyethanol is 2.5 M, and the author used a solution concentration of 1-2 M to grow thin monocrystals. Why did the authors not choose a concentration greater than 2 M to grow monocrystals?

Response: We would like to thank the reviewer’s comments. We employed a solution with slightly lower than the up limit, primarily due to the following reason. For the control system, the heating rate is insufficient, approximately 2°C h⁻¹, and we suggest using a faster heating rate (4°C h⁻¹) to shorten the heating process. This requires us to balance the competition between nucleation and growth. The solution-based crystal

growth model can be demonstrated using a dissolution-nucleation diagram¹⁷. As shown in Fig. R16, the diagram contains two curves corresponding to the solubility curve and the supersaturation curve. It is noteworthy that the supersaturation curve is determined through the process of nucleation experiment. Specifically, the nucleation temperatures of different concentrations of perovskite solutions were recorded under certain heating conditions, and the upper limit for different concentrations was regarded as the supersaturation curve. The two curves divide the entire region into three parts: the stable zone, growth zone, and nucleation zone. Above the supersaturation curve, nucleation will occur spontaneously once thermodynamic and kinetic requirements are satisfied. The steady-state nucleation rate (j_0) on the crystal surface can be expressed as¹⁸:

$$j_0 = A \exp\left(\frac{-\Delta G^*}{kT}\right) \quad (R1)$$

$$\Delta G^* = \Delta G_{Sol} - \Delta G_{Sur} \quad (R2)$$

$$\Delta G_{Sol} = -RT \ln(S) \quad (R3)$$

Where A represents the frequency factor of nucleation, ΔG^* is the free energy change and represents the activation energy for nucleation, k is the Boltzmann constant, T is the temperature, ΔG_{Sol} and ΔG_{Sur} are the Gibbs free energy change per unit bulk and surface, and S is saturation (the ratio of solute concentration to stable concentration).

Fig. R16 | The dissolution-nucleation diagram. The whole region is separated by the solubility curve and the supersaturation curve.

As the solution concentration increases, the system approaches supersaturation, making ΔG_{Sol} and ΔG^* more negative. This leads to an exponential increase in the nucleation rate. Actually, despite achieving efficient mass transfer in the HFG system, when using the 2.5 mol L^{-1} solution, the whole system will exhibit lots of unwanted nucleus due to the increased nucleation density. Based on the dissolution-nucleation diagram, the nucleation temperature interval for the 2 mol L^{-1} solution ranges from 55 to 60°C . Our studies indicate that there is no discernable impact on the final nucleation density and crystal quality by setting the initial HFG solution temperature at 50°C and then raising the temperature at a rate of 4°C h^{-1} , which further reduces the duration of experiment. To enhance the reader's comprehension, we introduced the discussion of equilibrium nucleation and growth in the revised manuscript.

Comment 10: 10. The author should provide the electron backscatter diffraction (EBSD) pattern with crystal boundaries (corresponding to Fig. 1f) in SI.

Response: We are grateful to the reviewer's constructive suggestions. As shown in Fig. R17, we have measured the electron backscatter diffraction pattern that contains grain boundaries and placed it in Fig. 1f. The pattern of HFG monocrystal demonstrates distinct grain boundaries and an individual crystallographic orientation.

Fig. R17 | EBSD mapping of the top surface of a HFG thin monocrystal.

Comment 11: 11. The author ought to provide a cross-section SEM images of the self-driven thin monocrystal device.

Response: Following the reviewer's suggestion, we added the cross-sectional SEM

images of the self-driven thin monocrystal device in Fig. R18, highlighting the configuration of ITO/PTAA/monocrystal/C60/BCP/Au, to enable readers to obtain more detailed structural information.

Fig. R18 | Cross-sectional SEM image of the self-driven thin monocrystal device.

Comment 12: 12. The thickness of the thin monocrystals (4-50 μm) in this work does not fulfill the requirements for TEM testing. The authors are requested to add the sample preparation process for TEM testing in the Structure characterization section.

Response: We thank the reviewer's comments. To prepare the TEM samples, we scraped thin monocrystals from the substrate, ground them into powder and finally dispersed them into toluene. Approximately 20 μL of the dispersion was then pipetted and drop-casted onto an ultrathin carbon-coated copper grid. The entire process was performed in a nitrogen glove box to eliminate any potential influence of air and moisture on the structure. We have detailed this process in the Methods section to give the reader a full understanding of the method we took to prepare TEM sample.

Comment 13: 13. In order to increase the standardization of the manuscript, it is recommended that the units be revised to a uniform format, such as mol/L should be revised to mol L⁻¹.

Response: We thank the reviewers for their careful examination of the manuscript. To ensure standardization, we modified the representation of units from 'mol/L' to 'mol L⁻¹'. Furthermore, we have carefully checked and revised the representation of all units in the revised manuscript.

Comment 14: 14. In this work the authors used the prepared monocrystal to record X-ray imaging. As we know, X-ray imaging for medical imaging, security check, and bioimaging, usually need large uniform film. Is it possible to present X-ray imaging over large area using the prepared monocrystal?

Response: Currently, commercial X-ray detectors are based on either the indirect or direct conversion method¹⁹. The indirect-conversion method uses a scintillator to convert X-rays into light, and then a photodetector to convert the light into electronic signals. Scintillators inevitably causes scattering of the luminance signal, impacting the image's spatial resolution²⁰. Furthermore, the additional energy conversion causes devices to have poor sensitivity. In contrast, the direct conversion method uses a photoconductor to convert X-rays directly into electronic signals, with simple instrument configurations able to optimize the conversion process to achieve higher resolution. Commercially available, large-area, flat-panel direct X-ray imaging systems are mainly based on amorphous selenium (α -Se), which can only be used for low sensitivity detection of soft X-ray (<30 keV) due to the absence of high Z value elements and poor charge transport properties²¹. Metal halide perovskite monocrystals featuring solution processable, high mobility-lifetime products, and large X-ray attenuation coefficients are promising candidates for the development of radiation detectors. The best perovskite X-ray detectors exhibit high sensitivities exceeding $5.2 \times 10^6 \mu\text{C Gy}^{-1} \text{cm}^{-2}$ under 1000 V cm^{-1} with the lowest detection limit well below that of α -Se (0.1 nGy s^{-1} vs. 5500 nGy s^{-1})^{22,23}, which completely meet the general medical imaging requirements.

Fig. R19 | **a** Photograph and **b** X-ray images of 5 mm thick aluminum plate mask plate with dolphin pattern. The X-ray images were obtained by the self-driven thin monocrystal device under is 540 nGy s^{-1} .

In our work, we developed a general approach to overcome synthetic limitation by using a high solute flux system, which is validated by the synthesis of 29 types of perovskites thin monocrystals. Combining X-Y scanning equipment, the HFG device with single-pixel enable X-ray imaging under the zero-bias mode. As a concept

demonstration, we further imaged a larger object, e.g., 110×50×5 mm aluminum plate mask plate with dolphin pattern. The obtained X-ray image exhibits excellent contrast (Fig. R19). In principle, expanding the range of motion of the X-Y scanning equipment can enable us to attain X-ray images over large-area ($>1 \text{ m}^2$).

Fig. R20 | Photograph of the pixel self-driven device.

Fig. R21 | The pixel current of self-driven device in dark and under X-ray irradiation.

Moreover, multi-pixel operating mode offers a promising route to practical X-ray imaging. An array detector was fabricated on monocrystal device with the configuration of ITO/PTAA/monocrystal/C60/BCP/Au, featuring a pixel size of $200 \times 200 \mu\text{m}$ (Fig. R20). The optoelectronic testing indicated that the dark current of the central 6×6 pixels ranged from 7.1–7.5 pA, while the photocurrent performance was between 469.7–475.9 pA, demonstrating excellent electrical uniformity (Fig. R21). This means that the monocrystals can be used not only for imaging by scanning, but also be pixelated into uniform 1D or 2D array for direct X-ray imaging. Once again, we greatly appreciate the reviewer's valuable suggestions.

Reviewer #3 (Remarks to the Author):

Comment 1: The paper presents a modified approach to developing perovskite monocrystals through a solvent composition modification method that enables improved growth rates, supports lower defect levels and can be adopted to a variety of compositions. The work itself builds on considerable foundations in monocrystal growth, which includes solvent and composition tunability. Through high level optoelectronic characterisation, it is shown that the synthetic approach yields significant improvements in mobility, lifetime and Mu tau product. This is capitalised through the fabrication of X-ray detectors in a vertical diode configuration and achieve very high sensitivity factors and low detection limits.

The work itself is interesting and presented at a high level. However, it bridges several scientific fronts with limited details in the key areas.

Response: We thank the reviewer's positive comments. Following the reviewer's suggestions, we have added a discussion of diffusion models and a detailed characterization of diffusion coefficients in terms of diffusion mechanisms. We also conducted a more in-depth analysis on the growth process of the crystal library, provided some universal concepts, and discussed the growth conditions for the refined design of some special crystals. Furthermore, in the section of X-ray detectors, some extended data, such as large-area imaging, pixel array uniformity, and device stability are introduced.

Comment 2: Firstly, the diffusion model presented is highly simplistic in its consideration and the discussion section around it is equally so. The fundamental nature of the precursors is suggested based on DLS, which has questionable validity at the sub nanometer level. No examination of the actual speciation has been conducted from a chemistry perspective, making this entire section highly speculative. I would think that comparative NMR would for instance provide an ability to determine diffusion in a much more accurate way. The consideration between the interplay of diffusion and reaction limits is not well presented or considered, and certainly not compared between precursors.

Response: We would like to thank the reviewer's comments. To respond to the comments, we summarize the questions and answer them as follows:

(1) The discussion about the diffusion model.

To the best of our knowledge, studies on the growth of perovskite single crystals

commonly rely on formulas to qualitative explain the experiment result, with few modeling analyses conducted. In our research, we built a classical two-step growth model for the quantitative analysis of the solution environment surrounding the monocrystal, which provides a guide for understanding the growth process. We acknowledge that the model is used as a basic consideration to study the crystal growth and diffusion behavior of perovskite monomers.

Fig. R22 | Simulated concentration field of solute near a MAPbI_3 monocrystal with varied substrate sizes. The monocrystal size grow from $150 \mu\text{m}$ to a certain size, i.e., **a** $300 \mu\text{m}$, **b** 1 mm , **c** 5 mm , and **d** 1 cm . The diffusion coefficient is $5 \times 10^{-10} \text{ m}^2 \text{ s}^{-1}$ and the initial solution concentration is 2.5 mol L^{-1} .

According to the reviewer's suggestions, we consider the influence of the substrate size, e.g., diffusion limit, on mass transfer of the system. Four different sizes ($300 \mu\text{m}$, 1 mm , 5 mm , and 10 mm) were selected for analysis, with the $150 \mu\text{m}$ seed crystal as the starting-point crystal (Fig. R22). All parameters in the model take the HFG system as an example, i.e. the diffusion coefficient is $5 \times 10^{-10} \text{ m}^2 \text{ s}^{-1}$ and the initial solution concentration is 2.5 mol L^{-1} . Note that the introduction of seed crystal does not affect the nucleation density of the entire system, as monomers tend to condense on the surface of the seed crystal rather than form new nuclei. We found that the diffusion layer might vary with reaction boundary size, which is significant in small containers,

like those smaller than 1.5 mm in Fig. R22a. However, the length of diffusion layer would be the same once the reaction boundary size reaches certain values (~1.5 mm for a crystal of 300 μm ; ~4 mm for a crystal of 1 mm; ~20 mm for a crystal of 5 mm; ~4 cm for a crystal of 10 mm). Considering the real experiment condition in which large enough substrate was employed with very few nucleus, the simulation under the assumption of unlimited reaction boundary is reasonable. This also emphasizes the critical role of diffusion step, as the rate-limiting process, in the thin monocrystal growth and presents a potential guideline to tailor the mass production.

In addition, we reconsider the assumption of the diffusion model that diffusion is the key factor dominating the growth process of monocrystal and add the corresponding discussion of the limits of diffusion and reactions. Crystal growth can be described by a two-step growth mode, which consists of the solute diffusion and the condensation of monomers on crystal surface. If solute diffusion limits the crystal growth rate, which is the favorable condition to obtain high-quality single crystals. On the other hand, if the monomers condensation rate is slower than the diffusion rate, the condensation rate becomes the limiting factor for crystal growth. Excess solute then leads to the formation of defect structures¹⁸. To quantitatively compare the criticality of solute diffusion and surface reaction, an effective factor η is defined as the ratio of the measured overall growth rate of the crystal to the growth rate when the crystal exposed to bulk solution. As the surface reaction becomes less important, $\eta \rightarrow 0$, the solute diffusion step dominates the whole growth process²⁴. Obviously, the growth rate of thin monocrystals using the space-confined method is significantly smaller than that of bulk monocrystal in free solution, the latter yielding millimeter-size monocrystals within minutes²⁵. Therefore, we believe that the diffusion step is the key factor dominating the growth of thin monocrystals (corresponding to an extremely small η), rather than surface reaction, which also support the conclusion of our simulations.

(2) Determination of the diffusion coefficient.

We thank the reviewer's constructive feedback, which has given us a new direction for systematically evaluating the diffusion of perovskite colloid diffusion. Diffusion-ordered spectroscopy (DOSY) is an exciting NMR technique used to probe the diffusion behavior of molecules in solution. Note the deuterated reagent is required to lock the magnetic field for NMR measurements, we attempted the commonly used deuterated reagents such as benzene, acetonitrile (ACN), chloroform, DMSO, etc. into the system. We found that most of the deuterated solvents cannot be compatible with

the perovskite precursor solution, while only the addition of ACN at a volume ratio of 10% have an insignificant impact on the colloid nature. Previous reports have demonstrated that the ACN does not affect the coordination state in the 2-ME and GBL systems through UV-vis absorption spectra²⁶. Furthermore, our DLS measurements show that the colloid size of 2-ME system is retained to be about 0.7 nm. We observed a slight reduction in the colloid size in GBL system (Fig. R23), which we attribute to the dissociation of the lead-iodine complex²⁷. Concerning the inconspicuous effect of ACN on colloidal chemistry, we operated DOSY measurements on these systems.

Fig. R23 | Colloidal size of **a** 2-ME and **b** GBL solution after introducing deuterated ACN at a volume ratio of 10%.

NMR results in Fig. R24 reveals a bigger diffusion coefficient in the HFG system, up to $3.0 \times 10^{-9} \text{ m}^2 \text{ s}^{-1}$, which is higher than $1.9 \times 10^{-9} \text{ m}^2 \text{ s}^{-1}$ in the control system. Because of the introduction of deuterated ACN, the colloid size of the control system becomes smaller and therefore may give rise to over-estimate of diffusion coefficient by NMR measurements.

Fig. R24 | The diffusion coefficient spectra of control and HFG system via NMR DOSY measurements.

For the dynamic light scattering technique, it determines particle size distribution in real time through fluctuations in elastic Rayleigh scattering caused by the individual Brownian motion of nanoparticles in solvent. For Rayleigh scattering, the intensity of the scattered light scales with d^6 , where d is the diameter of a particle. Reported studies indicate that the colloidal particle size of the perovskite precursor solution, as measured by DLS technology, ranges from 0.5–10 nm²⁸. In addition, the colloid size of perovskite also be proven to be 1–2 nm through cryo-electron microscopy²⁹, small angle neutron scattering and spin–echo small angle neutron scattering³⁰, which is consistent with the value measured by DLS. The colloid size of the perovskite precursor is determined by additives³¹, the type of solvent³² and pH value²⁸. For instance, by introducing polymer to the FAPbI₃ precursor, Ma *et al.* obtained perovskite colloids with size as low as 0.8 nm³¹.

Table R2 | Summary of the diffusion coefficient of perovskite solution measured by testing methods.

Measurement	Control	HFG
DLS	$1.81 \times 10^{-9} \text{ m}^2 \text{ s}^{-1}$	$5.55 \times 10^{-9} \text{ m}^2 \text{ s}^{-1}$
UV–vis	$1.7 \times 10^{-10} \text{ m}^2 \text{ s}^{-1}$	$5.4 \times 10^{-10} \text{ m}^2 \text{ s}^{-1}$
DOSY	$1.9 \times 10^{-9} \text{ m}^2 \text{ s}^{-1}$	$3.0 \times 10^{-9} \text{ m}^2 \text{ s}^{-1}$

We have summarized the diffusion coefficient in Table R2. We believe that the measured colloid size can be used to qualitatively characterize the relative sizes of the colloid sizes of HFG and control systems. This is also a well-established method to evaluate the diffusion coefficient of macromolecules and polymers^{33,34}. On the other hand, the DOSY measurements give the diffusion coefficient of the methylamine molecules in the solution system, whereas the absorption measurement gives the diffusion coefficient of the lead-iodine complex, resulting in a large difference between the two values. Anyhow, the DLS, UV–vis and NMR results jointly confirm the fast diffusion of lead iodide and methylamine in the HFG system in our work.

Comment 3: The use of the so called high flux growth must be highly dependent on the interplay between the solvent and the precursor chemistry. However, this is not explored, nor are any of the comparative growth considerations across the compositions presented. What is shown, namely the bandgaps vs temperature and optical images, should be expanded to provide depth to the comparison, for instance, growth rates, stability factors, structure etc. Through this comparison, one can make a more

generalised growth mechanism. With the data presented, there is insufficient evidence to validate this, merely to show that the approach works.

Response: We appreciate the constructive suggestions from the reviewer. To respond to the comments, we summarize the questions and answer them as follows:

(1) The perovskite precursor chemistry

The in-depth understanding of solvent chemistry could benefit the processing of perovskite materials and the development of their devices. Upon dissolution, perovskites dissociate into various species, such as Pb^{2+} , X^- inorganic ions, A^+ organic cations, and $[\text{PbX}_n]^{2-n}$ complexes³⁵, as confirm by the UV–vis absorption spectra (Fig. R25). The MAPbI_3 solution exhibits similar lead halide characteristic peaks, specifically Pb^{2+} , PbI_2 , and PbI_3^- .

Fig. R25 | UV–vis absorption spectra of the MAPbI_3 precursor solutions.

The strength between the precursor-solvent interaction can be semiquantitatively described by the donor number (D_N). Taking MAPbI_3 as an example, GBL and 2-ME with D_N of 17.8 and 19.7 can only dissolve lead iodide in the presence of MAI, but not alone^{26,27}. The distinct wavenumber blueshift in FT-IR spectra proves the existence of Pb-O bond (Fig. R26). During the temperature increase in the reverse temperature crystallization method, MA and solvent molecules compete for coordinating with Pb^{2+} , and the lead complex tends to combine with methylamine, which leads to crystal precipitation²⁶.

Fig. R26 | FT-IR spectra of **a** HFG and **b** control perovskite precursor solutions. The shift of characteristic peaks reveals the existence of hydrogen bonds and coordination bonds in perovskite precursor solution.

In addition, MA can establish robust hydrogen bonding interactions between the N-H groups and hydroxyl groups present in alcohols. The Kamlet-Taft β value has been demonstrated to indicate the strength of a hydrogen bond, whereby a higher β value corresponds to a stronger bond³⁶. In our work, we have verified the presence of hydrogen bonding in control and HFG system using FT-IR spectra (Fig. R26). We found that 2-ME exhibits a higher β value than GBL, which corresponds to stronger hydrogen bonding. This should contribute to the higher solubility of 2-ME for MAPbI₃.

To further investigate the atomic colloidal chemistry of perovskite precursors, we have performed *ab initio* molecular dynamics (AIMD) simulations of perovskite precursor species in different solvents at 300 K. The 2-ME molecules coordinates with lead iodide species by forming C-O-Pb. In comparison, GBL bonds with C=O-Pb. As shown in Fig. R27, the typical bond length of Pb-O is 2.0–4.0 Å, where the integrated distribution of the HFG system demonstrates the larger intensity than that of the control system, illustrating the slightly shorter Pb-O bond in HFG system.

Fig. R27 | **a**, Pb–O radial distribution functions ($g^{\text{Pb-O}}$) and **b** the integrated distribution of the HFG and control model systems, averaged over the 10 ps.

Fig. R28 | **a**, Pb L₃-edge XANES spectra of control and HFG perovskite solution and reference Pb foil and PbI₂ powder. The inset shows the magnified image. **b**, Fourier-transformed spectra of Pb L₃-edge EXAFS spectra of control and HFG perovskite solution.

To gain deeper insights into the electronic structure and local coordination environment of Pb species, X-ray absorption near edge structure (XANES) and extended X-ray absorption fine structure (EXAFS) analyses were conducted. In the R -space fourier-transformed EXAFS (Fig. 1g), both the control and HFG sample show two peaks around 2.15 Å and 2.67 Å, attributed to the scattering path of Pb-O and Pb-I bond, respectively³⁷. Both bonds were found to be shortened in the HFG system, which again support the results derived from the AIMD simulation. Concerning the higher donor

number of 2-ME, this solvent is thought to better solvate perovskite precursors, and further contribute to smaller colloid.

(2) Influence of composition on monocrystal growth

To investigate the effect of composition on crystallization, we summarize the growth parameter of perovskite monocrystals in terms of structure, growth rate, and temperature in Table R3. By comparing the growth rates of different monocrystals (Fig. R29), we obtained the following universal rules.

Fig. R29 | Growth rate of the HFG perovskite thin monocrystals along radial direction.

Fig. R30 | Growth rate per area of HFG perovskite thin monocrystals.

(1) For the A site of the perovskite structure, MA based perovskites always exhibit the most rapid crystal growth, followed by FA and Cs based ones. In particular, perovskites with larger cations such as BA and PEA exhibit comparatively slower growth rates. While the anisotropically grown $\text{PEA}_2\text{PbBr}_4$ monocrystal demonstrates the highest growth rate, its growth rate per area is comparatively slower (Fig. R30). It is concluded that the positive correlation between crystal growth rate and cation size, due to different diffusion rates.

(2) For different types of metal cations, perovskite monocrystals based on Pb, Bi, and Sb are relatively straightforward to synthesis with high growth rate. These perovskites are relatively stable as reported. Sn-based and Ge-based perovskites demonstrate relatively slow growth rates probably due to their inferior intrinsic stability^{38,39}. For halide double perovskite monocrystal, due to the unique structure of alternating monovalent and trivalent atoms^{40,41}, the lower growth rates are also observed.

(3) In the terms of monocrystal growth temperature, the monocrystals mentioned in this work can be obtained by a reasonable heating process below 70 °C. This is attributed to the high flux of monomers during the growth process in the HFG system. Only for some monocrystals with thermodynamically unstable phases (such as FA cation), relatively high temperature (90 °C) is required to avoid the formation of non-perovskite phases.

(4) We also checked the space group of all grown perovskites, and found it does not affect the growth rate or temperature of thin monocrystals in this work.

The growth parameters and corresponding discussion have been added in the revised supplementary information.

Table R3 | Summary of the growth parameters of thin monocrystals based on 2-ME solvent.

Type	Perovskite	Space group	Ramp rate	Temp./Time	Growth rate ($\mu\text{m h}^{-1}$)
3D	FA _{0.5} MA _{0.5} PbI ₃	/	10 °C h ⁻¹	90 °C/24 h	/
	Cs _{0.05} MA _{0.95} PbI ₃	Tetragonal (I4mcm)	5 °C h ⁻¹	90 °C/24 h	284.5
	Cs _{0.02} FA _{0.2} MA _{0.78} PbI ₃	Tetragonal (I4mcm)	5 °C h ⁻¹	90 °C/24 h	/
	MAPbI _{2.8} Br _{0.2}	Tetragonal (I4mcm)	5 °C h ⁻¹	70 °C/24 h	/
	MASnI ₃	Cubic (Pm-3m)	5 °C h ⁻¹	60 °C/12 h	15.4
	FASnI ₃	Cubic (Pm-3m)	5 °C h ⁻¹	60 °C/12 h	/
	MAGeI ₃	Trigonal (R3m)	5 °C h ⁻¹	40 °C/12 h	25.2
	FAGeI ₃	Trigonal (R3m)	5 °C h ⁻¹	40 °C/12 h	/
	CsGeI ₃	Trigonal (R3m)	5 °C h ⁻¹	40 °C/12 h	/
	MA ₂ AgSbI ₆	Orthorhombic	5 °C h ⁻¹	60 °C/3 h	30.1
	MA ₂ AgBiI ₆	Orthorhombic	5 °C h ⁻¹	60 °C/3 h	29.1
	Cs ₂ AgBiBr ₆	Cubic (Fm-3m)	5 °C h ⁻¹	60 °C/1 h	/
	2D (quasi 2D)	BA ₂ PbI ₄	Orthorhombic (Pbca)	10 °C h ⁻¹	80 °C/24 h
PEA ₂ PbI ₄		Triclinic (P-1)	10 °C h ⁻¹	70 °C/24 h	32.5
BDAPbI ₄		Triclinic (P1)	10 °C h ⁻¹	70 °C/12 h	/
PEA ₂ PbBr ₄		Triclinic (P-1)	5 °C h ⁻¹	40 °C/0.5 h	1632.8
	BA ₂ MAPb ₂ I ₇	Orthorhombic (Cc2m)	5 °C h ⁻¹	80 °C/36 h	24.8

Table R3 (continued) | Summary of the growth parameters of thin monocrystals based on 2-ME solvent.

Type	Perovskite	Space group	Ramp rate	Temp./Time	Growth rate ($\mu\text{m h}^{-1}$)
1D	DMAPbI ₃	Hexagonal (P6₃/mmc)	10 °C h ⁻¹	70 °C/12 h	/
	Cs ₃ Sb ₂ I ₉	Hexagonal (P6₃/mmc)	5 °C h ⁻¹	60 °C/0.5 h	157.7
	MA ₃ Sb ₂ I ₉	Hexagonal (P6₃/mmc)	5 °C h ⁻¹	60 °C/0.5 h	198.3
	Cs ₃ Bi ₂ I ₉	Hexagonal (P6₃/mmc)	5 °C h ⁻¹	60 °C/1 h	87.5
	MA ₃ Bi ₂ I ₉	Hexagonal (P6₃/mmc)	5 °C h ⁻¹	60 °C/0.5 h	239.5
	FA ₃ Bi ₂ I ₉	Hexagonal (P6₃mc)	5 °C h ⁻¹	60 °C/0.5 h	150.6
	Cs ₃ Cu ₂ I ₅	Orthorhombic (Pnma)	10 °C h ⁻¹	50 °C/3 h	37.2

Comment 4: The work on the optoelectronic properties is well conducted and presented. Minor points around this: details of the exact crystal sizes nor how the composition was determined aren't clearly presented.

Response: We thank the reviewer's valuable suggestions. For general photoelectric performance characterization and X-ray detection, we use devices with an electrode area of 400×400 μm . For SCLC and TOF testing, we used 100 μm thick devices due to the extremely low trap filling voltage and faster carrier transport of monocrystal. In addition, we adopted an electrode area of 200×200 to achieve high-precision imaging with a step size of 250 μm .

According to previous research on mixed cationic perovskite monocrystals, the ratio of the final material was measured by inductively coupled plasma mass spectrometry (ICP-MS) and ¹H nuclear magnetic resonance (NMR) spectroscopy. The ICP-MS results reveal the ratio of Cs to Pb to be 0.023:1 (32 $\mu\text{g L}^{-1}$ vs 2.1 mg L^{-1}). Then, we obtained the ratio of FA and MA to be 0.26:1 through NMR (Fig. R31). After calculation, the real composition of our final material is Cs_{0.023}FA_{0.202}MA_{0.775}PbI₃, which is

simplified to $\text{Cs}_{0.02}\text{FA}_{0.20}\text{MA}_{0.78}\text{PbI}_3$. Following the reviewer's suggestions, we added the description of the crystal composition in the revised manuscript, which provides more reliability for the experimental results.

Fig. R31 | ^1H NMR spectroscopy of the triple-cation perovskite monocrystal (DMSO-D_6 , 500 MHz).

Comment 5: The x-ray detector data suggests impressive device performance values. However, the devices weren't actually shown, and it would be useful to better understand what the effective area was of the devices and how they were actually fabricated. For instance, you used a shadow mask to deposit onto a 400-500 micron size pixel? How did you avoid shorting? Statistic data is not show here, and it would be useful to get a broader comparison to ensure that there aren't measurements errors.

Response: We would like to thank the reviewer's comments. We have shown a picture of the fabricated device, and its configuration is ITO/PTAA/monocrystal/C60/BCP/Au (Fig. R32). Nowadays, metal masks for device fabrication can be produced with minimum dimensions of a few micrometers⁴². We also shown the masks we used in Fig R33. Because of the large thickness of the perovskite layers, the device cannot be easily short-circuited during the fabrication. In addition, a homemade probe was used for the contact, and the other side was directly clamped on the ITO substrate to avoid short circuit.

Furthermore, we evaluated the sensitivity distribution of different self-driven devices, as shown in the Fig. R34. The sensitivity of the HFG monocrystal ranges from 1.30 to $1.74 \times 10^5 \mu\text{C Gy}^{-1} \text{cm}^{-2}$, which is higher than previously reported recorded values ($8.7 \times 10^4 \mu\text{C Gy}^{-1} \text{cm}^{-2}$). We also checked the uniformity of the device performance based on a 6×6 pixels array. It was fabricated on a monocrystal with the configuration

of ITO/PTAA/monocrystal/C60/BCP/Au, featuring a pixel size of $200 \times 200 \mu\text{m}$. The optoelectronic testing indicated that the dark current of all pixels ranged from $7.1\text{--}7.5 \text{ pA}$, while the photocurrent performance was between $469.7\text{--}475.9 \text{ pA}$, demonstrating excellent electrical uniformity (Fig. R35). These results further validate the reliability and consistency of our study.

Fig. R32 | Photograph of the pixel self-driven device.

Fig. R33 | Photograph of the mask.

Fig. R34 | The Sensitivity distribution of HFG monocrystal self-driven devices.

Fig. R35 | The pixel current of self-driven device in dark and under X-ray irradiation.

Comment 6: General remarks: It's not entirely clear what error margins are on the diffusion experiment presented in Fig 2. Further details around this should be included, particularly around the reproducibility and the impacts of the precursor addition methodology.

In general, the experimental errors are not presented across the paper.

Response: We thank the reviewer's comments. We divide the error of the entire diffusion experiment into the following reasons. The first one is the error of the spectrometer, and its absorbance resolution is 0.01. The absorbance of the final diffused solution is about 1, which have no significant impact from spectrometer. Secondly, the error is caused by solution addition. Specifically, a certain volume of solution is pipetted. Before injecting the cuvette, we need first immerse the pipette tip in corresponding pure solvent to eliminate the contribution to the absorbance of the liquid contaminated by the tip of the pipette. And we should carefully immerse the center of the bottom of the cuvette, inject the liquid completely into the bottom of the cuvette at a uniform rate and then slowly withdraw the pipette from the cuvette. We have added the details of the diffusion experiments to the Methods section. Furthermore, the diffusion experiment was repeated twice for control and HFG systems (Fig. R36). The error of each system is within the acceptable range. The HFG system always exhibits the rapid saturation of absorbance, which further adds to the credibility of the diffusion data.

Fig. R36 | Variation in time-dependent absorbance intensity of **a** control and **b** HFG perovskite precursors that diffuse from the bottom of cuvette.

Comment 7: The light based detector approach lacks any impact from my perspective and feel like a tack on. It would be more favourable to focus on the X-ray detector enhancements and statistics, rather than broader yet again into another section without completion of the others.

Response: We really appreciate the reviewer's valuable suggestions. We have removed the photodetector section and placed the application of HFG monocrystal entirely in X-ray detection. According to the suggestions of the reviewers, the research on the sensitivity distribution and uniformity of the pixel array of self-driven monocrystal device is supplemented. Combining X-Y scanning equipment, we further imaged a larger object, e.g., $110 \times 50 \times 5$ mm aluminum plate mask plate with dolphin pattern. The obtained X-ray image exhibits excellent contrast (Fig. R37). Moreover, as shown in Fig. R38, we have conducted an air stability study of the self-driven device without encapsulation under ambient atmosphere ($\text{RH} = 25 \pm 5\%$). After 600 h, the response current of HFG self-driven device retained 95 % of the initial value.

Fig. R37 | **a** Photograph and **b** X-ray images of 5 mm thick aluminum plate mask plate with dolphin pattern. The X-ray images were obtained by the self-driven thin monocrystal device under is 540 nGy s^{-1} .

Fig. R38 | Response current evolution of the unencapsulated HFG self-driven device under ambient atmosphere ($\text{RH} = 20 \pm 5\%$).

Reference

1. Sprau, C. *et al.* Highly efficient polymer solar cells cast from non-halogenated xylene/anisaldehyde solution. *Energy Environ. Sci.* **8**, 2744–2752 (2015).
2. Shi, X., Li, C., Guo, H. & Shen, S. Density, viscosity, and excess properties of binary mixtures of 2-(methylamino)ethanol with 2-methoxyethanol, 2-ethoxyethanol, and 2-butoxyethanol from 293.15 to 353.15 K. *J. Chem. Eng. Data* **64**, 3960–3970 (2019).
3. Liu, Y. *et al.* Thinness- and shape-controlled growth for ultrathin single-crystalline perovskite wafers for mass production of superior photoelectronic devices. *Adv. Mater.*, **28**, 9204–9209 (2016).
4. Chen, W. *et al.* Highly bright and stable single-crystal perovskite light-emitting diodes. *Nat. Photon.* **17**, 401–407 (2023).
5. Sakhatskyi, K. *et al.* Stable perovskite single-crystal X-ray imaging detectors with single-photon sensitivity. *Nat. Photon.* **17**, 510–517 (2023).
6. Chen, L. *et al.* Intrinsic phase stability and inherent bandgap of formamidinium lead triiodide perovskite single crystals. *Angew. Chem. Int. Ed.* **61**, e202212700 (2022).
7. Chen, Z. *et al.* Thin single crystal perovskite solar cells to harvest below-bandgap light absorption. *Nat. Commun.* **8**, 1890 (2017).
8. Deng, Y.H., Yang, Z.Q. & Ma, R.M. Growth of centimeter-scale perovskite single-crystalline thin film via surface engineering. *Nano Converg.* **7**, 25 (2020).
9. Schrader M. E. Young-dupre revisited. *Langmuir* **11**, 3585–3589 (1995).
10. Lei, Y. *et al.* A fabrication process for flexible single-crystal perovskite devices. *Nature* **583**, 790–795, (2020).
11. Liu, Y. *et al.* Thinness- and shape-controlled growth for ultrathin single-crystalline perovskite wafers for mass production of superior photoelectronic devices. *Adv. Mater.* **28**, 9204–9209, (2016).
12. Bao, C. *et al.* Low-noise and large-linear-dynamic-range photodetectors based on hybrid-perovskite thin-single-crystals. *Adv. Mater.* **29**, 1703209, (2017).
13. Yu, W. *et al.* Single crystal hybrid perovskite field-effect transistors. *Nat. Commun.* **9**, 5354, (2018).
14. Turedi, B. *et al.* Single-crystal perovskite solar cells exhibit close to half a millimeter electron-diffusion length. *Adv. Mater.* **34**, 2202390, (2022).

15. Guo, X. *et al.* Mitigating surface deficiencies of perovskite single crystals enables efficient solar cells with enhanced moisture and reverse-bias stability. *Adv. Funct. Mater.* **33**, 2213995, (2023).
16. Song, Y. *et al.* Detector-grade perovskite single-crystal wafers via stress-free gel-confined solution growth targeting high-resolution ionizing radiation detection. *Light Sci. Appl.* **12**, 85, (2023).
17. Wang, W. *et al.* Electronic-grade high-quality perovskite single crystals by a steady self-supply solution growth for high-performance X-ray detectors. *Adv. Mater.* **32**, 2001540 (2020).
18. Liu, Y. *et al.* Low-temperature-gradient crystallization for multi-inch high-quality perovskite single crystals for record performance photodetectors. *Mater. Today* **22**, 67–75 (2019).
19. Xu, X. *et al.* Halide perovskites: A dark horse for direct X-ray imaging. *EcoMat.* **2**, e12064 (2020).
20. Cao, F. *et al.* Shining emitter in a stable host: Design of halide perovskite scintillators for X-ray imaging from commercial concept. *ACS Nano* **14**, 5183–5193 (2020).
21. Shrestha, S. *et al.* High-performance direct conversion X-ray detectors based on sintered hybrid lead triiodide perovskite wafers. *Nat. Photon.* **11**, 436–440 (2017).
22. Song, Y. *et al.* Elimination of interfacial-electrochemical-reaction-induced polarization in perovskite single crystals for ultrasensitive and stable X-ray detector arrays. *Adv. Mater.* **33**, 2103078 (2021).
23. Kasap, S.O. X-ray sensitivity of photoconductors: application to stabilized a-Se. *J. Phys. D Appl. Phys.* **33**, 2853–2865 (2000).
24. Karpiński, P. H. Importance of the two-step crystal growth model. *Chem. Eng. Sci.* **40**, 641–646 (1985).
25. Saidaminov, M., *et al.* High-quality bulk hybrid perovskite single crystals within minutes by inverse temperature crystallization. *Nat. Commun.* **6**, 7586 (2015).
26. Deng, Y. *et al.* Tailoring solvent coordination for high-speed, room-temperature blading of perovskite photovoltaic films. *Sci. Adv.* **5**, eaax7537 (2019).
27. Chao, L. *et al.* Solvent engineering of the precursor solution toward large-area production of perovskite solar cells. *Adv. Mater.* **33**, 2005410 (2021).
28. Noel, N. K. *et al.* Unveiling the influence of pH on the crystallization of hybrid perovskites, delivering low voltage loss photovoltaics. *Joule* **1**, 328–343 (2017).
29. Dutta, N. S., Noel, N. K. & Arnold, C. B. Crystalline nature of colloids in

- methylammonium lead halide perovskite precursor inks revealed by cryo-electron microscopy. *J. Phys. Chem. Lett.* **11**, 5980–5986 (2020).
30. O’Kane, M. E. *et al.* Exploring nanoscale structure in perovskite precursor solutions using neutron and light scattering. *Chem. Mater.* **34**, 7232–7241 (2022).
 31. Ma, L. *et al.* A polymer controlled nucleation route towards the generalized growth of organic-inorganic perovskite single crystals. *Nat. Commun.* **12**, 2023 (2021).
 32. Zhang, H. *et al.* A universal co-solvent dilution strategy enables facile and cost-effective fabrication of perovskite photovoltaics. *Nat. Commun.* **13**, 89 (2022).
 33. Wasilewska, M., Adamczyk, Z. & Jachimska, B. Structure of fibrinogen in electrolyte solutions derived from dynamic light scattering (DLS) and viscosity measurements. *Langmuir* **25**, 3698–3704 (2009).
 34. Patil, S. M., Keire, D. A. & Chen, K. Comparison of NMR and dynamic light scattering for measuring diffusion coefficients of formulated insulin: implications for particle size distribution measurements in drug products. *AAPS J.* **19**, 1760–1766 (2017).
 35. Rahimnejad, S., Kovalenko, A., Forés, S. M., Aranda, C. & Guerrero, A. Coordination chemistry dictates the structural defects in lead halide perovskites. *ChemPhysChem.* **17**, 2795–2798, (2016).
 36. Huang, X. *et al.* Solvent gaming chemistry to control the quality of halide perovskite thin films for photovoltaics. *ACS Cent. Sci.* **8**, 1008–1016 (2022).
 37. Hui, W. *et al.* Stabilizing black-phase formamidinium perovskite formation at room temperature and high humidity. *Science* **371**, 1359–1364 (2021).
 38. Cao, J. & Yan, F. Recent progress in tin-based perovskite solar cells. *Energy Environ. Sci.* **14**, 1286–1325 (2021).
 39. Chiara, R., Morana, M. & Malavasi, L. Germanium-based halide perovskites: materials, properties, and applications. *ChemPlusChem* **86**, 879–888 (2021).
 40. Shadabroo, M. S., Abdizadeh, H. & Golobostanfard, M. R. Elpasolite structures based on A₂AgBiX₆ (A: MA, Cs, X: I, Br): Application in double perovskite solar cells. *Mater. Sci. Semicond. Process.* **125**, 105639 (2021).
 41. Ghasemi, M. *et al.* Lead-free metal-halide double perovskites: from optoelectronic properties to applications *Nanophotonics* **10**, 2181–2219 (2020)
 42. Zhang, B. *et al.* 90% yield production of polymer nano-memristor for in-memory computing. *Nat. Commun.* **12**, 1984 (2021).

REVIEWER COMMENTS

Reviewer #1 (Remarks to the Author):

In the revised version of the manuscript, the authors have addressed most of the questions. However, there are still some issues that should be addressed.

1. The author should provide the crystal growth rate for other glycol ether solvents system to demonstrate their effectiveness.

2. The authors only show the growth of a monocrystal in the in-situ video (visual field range, ca. $5 \times 5 \text{ mm}^2$). The side around the monocrystal is clean. Is there other random nucleation on the substrate? The authors should provide an in-situ video from the beginning of the crystalization over a large area or the entire substrate.

3. In previous reports, γ -butyrolactone, DMSO and DMF were all able to grow thin monocrystals of millimeter size (Nature Communications, 2018, 9, 5302., Angew. Chem. Int. Ed. 2021, 60, 2629.). In this work the sizes of monocrystals are all in the hundred-micron range (Fig.3a). To demonstrate the advantage of this work, the various perovskite monocrystals with larger size should be presented.

Reviewer #1 (Remarks to the Author):

Comment 1: In the revised version of the manuscript, the authors have addressed most of the questions. However, there are still some issues that should be addressed.

1. The author should provide the crystal growth rate for other glycol ether solvents system to demonstrate their effectiveness.

Response: We thank the reviewer for the insightful comments on our manuscript. In response to the reviewer's suggestion, the crystal growth rates have been measured and compared to those presented in the original manuscript. As shown in Fig. R1, most of the glycol ether solvents exhibit a significantly higher growth rate compared to the GBL (approximately twofold higher). It is noteworthy that the diffusion coefficients of 3-ME and GBL are very close ($2.3 \times 10^{-10} \text{ m}^2 \text{ s}^{-1}$ vs $1.7 \times 10^{-10} \text{ m}^2 \text{ s}^{-1}$), further illustrating their similar performance in terms of crystal growth rate. We attribute this variation to the increment of chain length and solvent viscosity of 3-ME, which results in the reduction in mass transfer. This addition provides a more comprehensive understanding of the effectiveness of glycol ether solvents systems in our study. The growth rate data for the additional glycol ether solvent systems have been included in the revised manuscript.

Fig. R1 | Growth rate of the MAPbI₃ monocrystals based on different solvent systems.

Comment 2: 2. The authors only show the growth of a monocrystal in the in-situ video (visual field range, ca. $5 \times 5 \text{ mm}^2$). The side around the monocrystal is clean. Is there other random nucleation on the substrate? The authors should provide an in-situ video from the beginning of the crystallization over a large area or the entire substrate.

Response: We are grateful to the reviewer's constructive suggestions. In our previous response, we presented a video that showed the microscopic growth process of a monocrystal, accompanied by an in-depth examination of its growth behavior. Here, we conducted further experiments and captured in-situ videos that cover a larger area (4.87×8.65 cm) from the beginning of the crystallization process. This allows us to observe any potential random nucleation events on the substrate, providing a more thorough understanding of the crystal growth dynamics. In fact, random nucleation may occur across the entire visual field; however, the overall nucleation density remains relatively low during the entire process. Only few nuclei can be observed in the central region of the substrate, which finally grow to over centimeter scale at 28 hours.

Comment 3: 3. In previous reports, γ -butyrolactone, DMSO and DMF were all able to grow thin monocrystals of millimeter size (Nature Communications, 2018, 9, 5302., Angew. Chem. Int. Ed. 2021, 60, 2629.). In this work the sizes of monocrystals are all in the hundred-micron range (Fig.3a). To demonstrate the advantage of this work, the various perovskite monocrystals with larger size should be presented.

Response: We greatly appreciate the constructive suggestions from the reviewer. Following the reviewer's suggestion, we optimized the monocrystal growth process by extending the crystal growth time, optimizing the ramp rate, utilizing larger substrate sizes (see details in Table R1). We have successfully grown all crystals with the size of millimeter scale (Fig. R2). Also, we updated the growth rate data in Table R1, which is close to the data provided earlier. This achievement not only validates the superiority of using glycol ether solvents for monocrystal growth process, as proposed in this study, but also offers a broader range of potential models for practical optoelectronic device fabrication. Once again, we greatly appreciate the reviewer's valuable comments and suggestions.

Fig. R2 | Library of as-grown perovskite thin monocrystals via the high-flux approach. Optical images of 28 types of the HFG perovskite thin monocrystals. Scale bars: 1 mm.

Table R1 | Summary of the growth parameters of thin monocrystals based on 2-ME solvent.

Type	Perovskite	Space group	Ramp rate	Temp./Time	Growth rate ($\mu\text{m}^2 \text{h}^{-1}$)
3D	MAPbI _{2.8} Br _{0.2}	Tetragonal (I4mcm)	5 °C h ⁻¹	70 °C/48 h	3.60×10^5
	FA _{0.5} MA _{0.5} PbI ₃	/	8 °C h ⁻¹	90 °C/48 h	2.83×10^5
	Cs _{0.05} MA _{0.95} PbI ₃	Tetragonal (I4mcm)	5 °C h ⁻¹	90 °C/48 h	3.31×10^5
	Cs _{0.02} FA _{0.2} MA _{0.78} PbI ₃	Tetragonal (I4mcm)	5 °C h ⁻¹	90 °C/48 h	3.15×10^5
	MASnI ₃	Cubic (Pm-3m)	3 °C h ⁻¹	60 °C/24 h	2.73×10^5
	FASnI ₃	Cubic (Pm-3m)	3 °C h ⁻¹	60 °C/24 h	2.00×10^5
	MAGeI ₃	Trigonal (R3m)	3 °C h ⁻¹	40 °C/36 h	7.54×10^4
	FAGeI ₃	Trigonal (R3m)	3 °C h ⁻¹	40 °C/24 h	3.92×10^4
	CsGeI ₃	Trigonal (R3m)	3 °C h ⁻¹	40 °C/24 h	2.71×10^4
	MA ₂ AgSbI ₆	Orthorhombic	3 °C h ⁻¹	70 °C/48 h	9.96×10^4
	MA ₂ AgBiI ₆	Orthorhombic	3 °C h ⁻¹	70 °C/48 h	1.19×10^5
	Cs ₂ AgBiBr ₆	Cubic (Fm-3m)	3 °C h ⁻¹	60 °C/24 h	3.79×10^4
	2D (quasi 2D)	BA ₂ PbI ₄	Orthorhombic (Pbca)	8 °C h ⁻¹	80 °C/72 h
PEA ₂ PbI ₄		Triclinic (P-1)	8 °C h ⁻¹	70 °C/72 h	6.17×10^4
BDAPbI ₄		Triclinic (P1)	8 °C h ⁻¹	70 °C/72 h	5.89×10^4
PEA ₂ PbBr ₄		Triclinic (P-1)	3 °C h ⁻¹	40 °C/72 h	5.69×10^4
BA ₂ MAPb ₂ I ₇		Orthorhombic (Cc2m)	3 °C h ⁻¹	90 °C/96 h	1.44×10^5

Table R1 (continued) | Summary of the growth parameters of thin monocrystals based on 2-ME solvent.

Type	Perovskite	Space group	Ramp rate	Temp./Time	Growth rate ($\mu\text{m}^2 \text{h}^{-1}$)
1D	DMAPbI ₃	Hexagonal (P6₃/mmc)	10 °C h ⁻¹	70 °C/24 h	1.10 × 10 ⁵
		Hexagonal (P6₃/mmc)	3 °C h ⁻¹	60 °C/24 h	4.34 × 10 ⁴
		Hexagonal (P6₃/mmc)	3 °C h ⁻¹	60 °C/36 h	8.11 × 10 ⁴
		Hexagonal (P6₃/mmc)	3 °C h ⁻¹	60 °C/36 h	3.90 × 10 ⁴
		Hexagonal (P6₃/mmc)	3 °C h ⁻¹	60 °C/36 h	1.03 × 10 ⁵
		Hexagonal (P6₃mc)	3 °C h ⁻¹	60 °C/36 h	8.33 × 10 ⁴
		Orthorhombic (Pnma)	10 °C h ⁻¹	50 °C/36 h	1.56 × 10 ⁴

Table R2 | Summary of the growth parameters of thin monocrystals based on the mixed solvent.

Perovskite	Solvent	Space group	Ramp rate	Temp./Time	Growth rate ($\mu\text{m}^2 \text{h}^{-1}$)
FAPbBr ₃	DMF/GBL/2-ME 1:1:1 (volume ratio)	Cubic (Pm-3m)	4 °C/h	50 °C/48 h	5.26 × 10 ⁴
MAPbBr ₃	DMF/2-ME 1:1 (volume ratio)	Cubic (Pm-3m)	4 °C/h	50 °C/36 h	7.01 × 10 ⁴
MAPbBr ₂ Cl	DMF/2-ME 4:1 (volume ratio)	Cubic (Pm-3m)	4 °C/h	40 °C/48 h	3.51 × 10 ⁴
MAPbCl ₃	DMSO/2-ME 1:1 (volume ratio)	Cubic (Pm-3m)	4 °C/h	50 °C/48 h	2.64 × 10 ⁴

REVIEWERS' COMMENTS

Reviewer #1 (Remarks to the Author):

The revised manuscript addresses our comments, which reinforces the significance of the research findings and contributes to a comprehensive understanding of a perovskite growth strategy based on high solute flux. The inclusion of the updated data and figures significantly enhances the manuscript's quality. Hence it can be considered for publication.